# WHY SO PESSIMISTIC? ESTIMATING UNCERTAINTIES FOR OFFLINE RL THROUGH ENSEMBLES, AND WHY THEIR INDEPENDENCE MATTERS

## ABSTRACT

In offline/batch reinforcement learning (RL), the predominant class of approaches with most success have been "support constraint" methods, where trained policies are encouraged to remain within the support of the provided offline dataset. However, support constraints correspond to an overly pessimistic assumption that actions outside the provided data may lead to worst-case outcomes. In this work, we aim to relax this assumption by obtaining uncertainty estimates for predicted action values, and acting conservatively with respect to a lower-confidence bound (LCB) on these estimates. Motivated by the success of ensembles for uncertainty estimation in supervised learning, we propose MSG, an offline RL method that employs an ensemble of independently updated Q-functions. First, theoretically, by referring to the literature on infinite-width neural networks, we demonstrate the crucial dependence of the quality of derived uncertainties on the manner in which ensembling is performed, a phenomenon that arises due to the dynamic programming nature of RL and overlooked by existing offline RL methods. Our theoretical predictions are corroborated by pedagogical examples on toy MDPs, as well as empirical comparisons in benchmark continuous control domains. In the significantly more challenging antmaze domains of the D4RL benchmark, MSG with deep ensembles by a wide margin surpasses highly well-tuned state-of-the-art methods. Consequently, we investigate whether efficient approximations can be similarly effective. We demonstrate that while some very efficient variants also outperform current state-of-the-art, they do not match the performance and robustness of MSG with deep ensembles. We hope that the significant impact of our less pessimistic approach engenders increased focus into uncertainty estimation techniques directed at RL, and engenders new efforts from the community of deep network uncertainty estimation researchers whom thus far have not employed offline reinforcement learning domains as a testbed for validating modern uncertainty estimation techniques.

## 1 INTRODUCTION

Offline reinforcement learning (RL), also referred to as batch RL, is the setting where we are provided with a dataset of interactions with a Markov Decision Process (MDP), and the goal is to learn an effective policy without further interactions with the MDP. Offline RL holds the promise of data-efficiency through data reuse and improved safety due to minimizing the need for policy rollouts. As a result, offline RL has been a subject of significant renewed interest in the machine learning literature (Levine et al., 2020).

One common approach to offline RL, known as model-free, uses value estimation through approximate dynamic programming (ADP). The predominant algorithmic philosophy with most success in ADP-based offline RL is to limit obtained policies to the support set of the available offline data, with the intuition being that such constraints would reduce inaccurate value estimates since the actions chosen by the policy are close to the observed data. A large variety of methods have been developed for enforcing such constraints, examples of which include regularizing policies with behavior cloning objectives (Kumar et al., 2019; Fujimoto & Gu, 2021), only performing updates on actions observed in (Peng et al., 2019; Nair et al., 2020; Wang et al., 2020; Ghasemipour et al., 2021)

or close to (Fujimoto et al., 2019) the offline dataset, and regularizing to lower the estimated value of actions not seen in the dataset (Wu et al., 2019; Kumar et al., 2020; Kostrikov et al., 2021).

Support constraint forms of regularization correspond to an overly pessimistic assumption that any action outside the provided data leads to highly negative outcomes (Buckman et al., 2020). Instead, it would be preferable if we could place more trust into the predictions of value networks beyond the training dataset. Indeed in adjacent fields of A.I., for different tasks of interest, researchers have spent significant effort developing architectures with built-in inductive biases to help neural networks not only obtain strong accuracy on the training set, but perform well beyond the data they were trained on: convolutional and residual networks for computer vision (He et al., 2016), recurrent networks and transformers for NLP (Vaswani et al., 2017), graph neural networks for graph-based data (Li et al., 2015), and Nerf and Siren for effective signal modelling (Sitzmann et al., 2020; Mildenhall et al., 2020).

However, trusting the predictions of neural networks beyond the training data is a challenging ordeal. Many works have demonstrated that neural networks are prone to making high-confidence incorrect predictions outside the training set, even on i.i.d. test data (Guo et al., 2017). This is a particularly major issue for value estimation, since errors rapidly accumulate through dynamic programming and lead to catastrophic results. Thus, in order to trust our models beyond the training set, a potentially promising approach would be to obtain high quality uncertainty estimates on predicted action values.

In current supervised learning literature, "Deep Ensembles" and their more efficient variants have been shown to be the most effective method for uncertainty estimation (Ovadia et al., 2019). In this work, we aim to transfer the success of ensembles for uncertainty estimation to the setting of offline RL. We begin by presenting MSG, an actor-critic offline RL algorithm leveraging an ensemble of Q-value functions. MSG trains $N$ Q-value functions completely independent of one-another, and updates an actor network with respect to a lower confidence bound (LCB) on action values obtained from the ensemble. By referring to the literature on infinite-width neural networks, we theoretically demonstrate the critical importance of independence in Q-functions, a deviation from standard practice in RL which more often uses the full ensemble to compute identical target values for training each ensemble member. Our theoretical predictions are corroborated by experiments in a toy MDP, and their relevance to practical settings is also verified through benchmark experiments.

In established benchmarks for offline RL, we demonstrate that MSG matches, and in the more challenging domains, significantly exceeds the prior state-of-the-art. Inspired by this success, we investigate whether the performance of MSG can be recovered through modern efficient approaches to ensembling. While we demonstrate that efficient ensembles continue to outperform current state-of-the-art batch RL algorithms, they cannot recover the performance of MSG using full deep ensembles.

**We hope that our work highlights some of the unique challenges of uncertainty estimation in reinforcement learning, and the significant impact it can have on the performance of algorithms.** We also hope that our work encourages stronger engagement from researchers specializing in uncertainty estimation techniques – who typically use computer vision and NLP tasks as benchmarks – to use offline RL as an additional testbed. Our results reveal that offline RL presents unique challenges not seen in standard supervised learning, due to the accumulation of errors over multiple back ups, and can be a valuable domain for testing novel uncertainty estimation techniques.

## 2 RELATED WORK

Uncertainty estimation is a core component of RL, since an agent only has a limited view into the mechanics of the environment through its available experience data. Traditionally, uncertainty estimation has been key to developing proper *exploration* strategies such as upper confidence bound (UCB) and Thompson sampling (Lattimore & Szepesvári, 2020), in which an agent is encouraged to seek out paths where its uncertainty is high. Offline RL presents an alternative paradigm, where the agent must act conservatively and is thus encouraged to seek out paths where its uncertainty is low (Buckman et al., 2020). In either case, proper and accurate estimation of uncertainties is paramount. To this end, much research has been produced towards the end of devising provably correct uncertainty estimates (Thomas et al., 2015; Feng et al., 2020; Dai et al., 2020), or, at least, bounds on uncertainty that are good enough for acting either exploratory (Strehl et al., 2009) or

conservatively (Kuzborskij et al., 2021). However, these approaches require exceedingly simple environment structure, typically either a finite discrete state and action space or linear spaces with linear dynamics and rewards.

While theoretical guarantees for uncertainty estimation are more limited in practical situations with deep neural network function approximators, a number of works have been able to achieve practical success, for example using deep network analogues for count-based uncertainty (Ostrovski et al., 2017), Bayesian uncertainty (Ghavamzadeh et al., 2016; Yang et al., 2020), and bootstrapping (Osband et al., 2019; Kostrikov & Nachum, 2020). Many of these methods employ ensembles. In fact, in continuous control RL, it is common to use an ensemble of two value functions and use their minimum for computing a target value during Bellman error minimization (Fujimoto et al., 2018). A number of works in offline RL have extended this to propose backing up minimums or lower confidence bound estimates over larger ensembles (Kumar et al., 2019; Wu et al., 2019; Agarwal et al., 2020). In our work, we continue to find that ensembles are extremely useful for acting conservatively, but the manner in which ensembles are used is critical. Specifically our proposed MSG algorithm advocates for using independently learned ensembles, without sharing of target values, and this import design decision is supported by empirical evidence.

The widespread success of ensembles for uncertainty estimation in RL echoes similar findings in supervised deep learning. While there exist proposals for more technical approaches to uncertainty estimation (Li et al., 2007; Neal, 2012; Pawlowski et al., 2017), ensembles have repeatedly been found to perform best empirically (Lee et al., 2015; Lakshminarayanan et al., 2016). Much of the active literature on ensembles in supervised learning is concerned with computational efficiency, with various proposals for reducing the compute or memory footprint of training and inference on large ensembles (Wen et al., 2020; Zhu et al., 2019; Havasi et al., 2020). While these approaches have been able to achieve impressive results in supervised learning, our empirical results suggest that their performance suffers significantly in an offline RL setting compared to deep ensembles, and even naive Multi-Head ensembles (Lee et al., 2015; Osband et al., 2016; Tran et al., 2020) which are not considered to be as effective in the supervised learning setting(Havasi et al., 2020).

## 3 METHODOLOGY

**Notation** Throughout this work, we represent Markov Decision Process (MDP) as $M = \langle \mathcal{S}, \mathcal{A}, r, \mathcal{P}, \gamma \rangle$, with state space $\mathcal{S}$, action space $\mathcal{A}$, reward function $r : \mathcal{S} \times \mathcal{A} \rightarrow \mathbb{R}$, transition dynamics $\mathcal{P}$, and discount $\gamma$. In offline RL, we assume access to a dataset of interactions with the MDP, which we will represent as collection of tuples $D = \{(s, a, s', r, term)\}^N$, where $t$ is an indicator variable that is set to True when $s'$ is a terminal state.

Two neural networks with an identical architecture, trained in an identical manner, will make different predictions outside the training set when their weights are initialized with different random draws from the initial weight distribution. So a natural question is, "Which network's predictions can we trust?" An answer to this question can be motivated by referring to the literature on infinite-width networks. When performing mean-squared error regression in the infinite-width regime, informally speaking, an intriguing property is that the distribution of predictions on unseen data is given by a Gaussian Process whose kernel function is solely defined by, 1) the architecture, and 2) the choice of initial weight distribution (Lee et al., 2019). Hence, to leverage the built-in architectural inductive biases of value networks for offline RL, we can train policies to act with respect to the lower-confidence bound (LCB) of the derived Gaussian Process.

In what follows, we present our proposed algorithm, MSG, which leverages ensembles to approximate the LCB. Additionally, using the literature on infinite-width networks, we demonstrate the theoretical advantage of forming ensembles in the manner we propose, which deviates from the current use of ensembles in the reinforcement learning literature.

### 3.1 MODEL STANDARD-DEVIATION GRADIENTS (MSG)

MSG follows an actor critic setup where in each iteration, we first estimate the Q-values of the current policy, and subsequently optimize the policy through gradient ascent on the lower confidence bound of action value estimates.

**Policy Evaluation** At the beginning of training, we create an ensemble of $N$ Q-functions by taking $N$ samples from the initial weight distribution. Throughout training, the loss for the Q-functions is the standard least-square Q-evaluation loss,

$$\mathcal{L}(\theta^i) = \mathbb{E}_{(s,a,r,s',t)\sim D}\Big[\big(Q_{\theta^i}(s,a) - y^i(r,s',term,\pi)\big)^2\Big] \tag{1}$$

$$y^i := r + (1 - term)\cdot\gamma\cdot\mathbb{E}_{a'\sim\pi(s')}\Big[Q_{\bar{\theta}^i}(s',a')\Big] \tag{2}$$

where $\theta^i, \bar{\theta}^i$ denote the parameters and target network parameters for the $i^{\text{th}}$ Q-function, and $term = \mathbb{1}[s'$ is terminal$]$. In practice, the expectation in equation 1 is estimated by a minibatch, and the expectation in equation 2 is estimated with a single action sample from the policy. After every update to the Q-function parameters, their corresponding target parameters are updated to be an exponential moving average of the parameters in the standard fashion. **A key factor to note is that, in contrast to the typical usage of ensembles in actor critic algorithms, each ensemble member's update (and $y^i$) is completely independent from other ensemble members. As we will present below, this an important algorithmic choice, both theoretically (Theorem 4.1) and empirically (section 5.3.2).**

**Policy Optimization** As described above, the choice of architecture and weight initialization induces a distribution on predictions. Having approximated this distribution using an ensemble of value networks, we can optimize the policy with respect to a lower confidence bound on action values. Specifically, our proposed policy optimization objective in MSG is,

$$\max_\pi \mathbb{E}_{s\sim D, a\sim\pi(s)}\Big[\mu(s,a) + \beta\cdot\sigma(s,a)\Big] \quad\text{where}\quad \mu(s,a) = \operatorname*{mean}_i\Big[Q_{\theta^i}(s,a)\Big], \quad \sigma(s,a) = \operatorname*{std}_i\Big[Q_{\theta^i}(s,a)\Big]$$

where $\beta \in \mathbb{R}$ is a hyperparameter that trades off conservatism and optimism. As our problem setting is that of offline RL requiring conservatism, we use $\beta \leq 0$.

### 3.2 THE TRADE-OFF BETWEEN TRUST AND PESSIMISM

While our hope is to leverage the distributions induced by the architecture choice, it is not always feasible to do so; designed architectures can still be fundamentally biased in some manner, or we can simply be in a setting with insufficient data coverage. Thus we need to trade-off trusting the generalization ability of our models with the pessimistic approach of support constraints. In this work, inspired by CQL (Kumar et al., 2020), we add the following regularizer to our policy evaluation objective (equation 1),

$$\mathcal{R}(\theta^i) = \mathbb{E}_{s\sim D}\Big[\mathbb{E}_{a\sim\pi(a|s)}[Q_{\theta^i}(s,a)] - Q_{\theta^i}(s, a_D)\Big] \tag{3}$$

where $a_D$ denotes the action taken in state $s$ in the dataset. We control the contribution of $\mathcal{R}(\theta^i)$ by weighting this term with weight parameter $\alpha$. Practically, we estimate the outer expectation using the states in the mini-batch, and we approximate the inner expectation using a single sample from the policy. An example scenario where we have observed such a regularizer is helpful is when the offline dataset only contains a narrow (e.g., expert) data distribution. We believe this is because the power of ensembles comes from predicting a value distribution for unseen $(s,a)$ based on the available training data. Thus, if no data for sub-optimal actions is present, ensembles cannot make accurate predictions and increased pessimism for unseen actions becomes necessary.

## 4 INDEPENDENCE IN ENSEMBLES MATTERS

### 4.1 THE STRUCTURE OF UNCERTAINTIES DEPENDS ON HOW ENSEMBLING IS DONE

In the reinforcement learning literature, ensembles are widely used for combatting over-estimation bias (Haarnoja et al., 2018; Fujimoto et al., 2018; 2019; Kumar et al., 2020; Agarwal et al., 2020; Ghasemipour et al., 2021). However, the typical usage of ensembles is to compute a target value $y$ using an ensemble of target networks, and subsequently update all the Q-functions with respect to the same target, typically the minimum over targets. Hence, the objectives for all the Q-functions share the same target value. In this section, drawing upon the literature of infinitely wide networks, we demonstrate that having the updates for ensemble members be independent of one another – as

done in MSG – results in uncertainty estimates that align more closely with intuitive expectations, compared to when their targets are shared.

We study this difference in the setting of policy evaluation using an infinite ensemble of infinite-width Q-function networks (i.e. an ensemble of infinitely many Q-functions, each of which is an infinite-width neural network, with the only difference being that their weights are initialized through independent random draws from the initial weight distribution). *For sake of simplicity of derivations we assume that the policy we are trying to evaluate is deterministic and that we do not have terminal states in the MDP.* The policy evaluation routine we consider is Fitted Q-Evaluation (Fonteneau et al., 2013), which can be described as repeatedly performing the following steps,

- **Compute TD Targets** For each $(s, a, r, s') \in D$ compute the TD targets $y^i(r, s', \pi)$
- **Fit the Q-functions** For each ensemble member, optimize the following objective until convergence using full batch gradient descent.

$$\frac{1}{|D|} \sum_{(s,a,r,s') \in D} \left[ \left( Q_{\theta^i}(s, a) - y^i(r, s', \pi) \right)^2 \right]. \tag{4}$$

First we establish some notation. Let $\mathcal{X}$ denote a matrix where the rows are the state-action pairs $(s, a)$ in the offline dataset. Let $R$ be the $|D| \times 1$ matrix containing the rewards observed after each $(s, a)$ in $\mathcal{X}$. Let $\mathcal{X}'$ denote a matrix where the rows are the next state and policy action $(s', \pi(s'))$ Additionally let,

$$\hat{\Theta}_0(A, B) := \nabla_\theta Q_\theta(A) \cdot \nabla_\theta Q_\theta(B)^T|_{t=0} \qquad \hat{\Theta}_0^{-1} := \hat{\Theta}_0(\mathcal{X}, \mathcal{X})^{-1} \qquad C := \hat{\Theta}_0(\mathcal{X}', \mathcal{X}) \cdot \hat{\Theta}_0^{-1}.$$

$\hat{\Theta}_0(A, B)$ above is referred to as the tangent kernel (Jacot et al., 2018), which is defined as the outerproduct of gradients of the Q-function, at initialization (iteration $t = 0$). The definition of $\hat{\Theta}_0$ does not contain the variable $i$ indexing the ensemble members because, at infinite width, $\hat{\Theta}_0(A, B)$ converges to a deterministic kernel and hence is the same for all ensemble members.

Intuitively, $C$ is a $|D| \times |D|$ matrix where $c_{p,q}$ (the element at row $p$, column $q$) captures a notion of similarity between $(s', \pi(s'))$ in the $p^{th}$ row of $\mathcal{X}'$, and $(s, a)$ in the $q^{th}$ row of $\mathcal{X}$. Let $\mathcal{Y}_t^i$ (with shape $|D| \times 1$) denote the targets used for fitting $Q_{\theta^i}$ at iteration $t$.

Here, we will study the difference between the following two methods for computing targets: one where each ensemble member uses its own predictions to compute the TD targets (analogous to MSG), and another where all ensemble members share same target (analogous to typical use of ensembles in offline RL):

- **Method 1 (MSG, Independent Targets):** $y^i(s, a) = r + \gamma \cdot Q_{\theta^i}(s', \pi(s))$

- **Method 2 (Shared LCB):** $\forall i, y^i(s, a) = r + \gamma \cdot \left[ \underset{ensemble}{\mathbb{E}} \left[ Q_{\theta^i}(s', \pi(s)) \right] - \underset{ensemble}{\mathrm{Std}} \left[ Q_{\theta^i}(s', \pi(s)) \right] \right]$

As can be seen in Appendix F, for the two methods considered above, under the described policy evaluation procedure, the values $Q_{\theta^i}(s, a)|_t$ (at iteration $t$) can be computed in closed form for all $(s, a) \in \mathcal{S} \times \mathcal{A}$. This enables us to compare the final distribution of $Q(s, a)$ after policy evaluation under two methods.

**Theorem 4.1.** *After $t$ iterations, for both Method 1 and Method 2 we have,*

***Independent:*** $\mathrm{LCB}\left( Q_{t+1}(\mathcal{X}') \right) \approx (1 + \ldots + \gamma^t C^t) CR - \sqrt{\mathbb{E}\left[ \left( (1 + \ldots + \gamma^t C^t)(Q_0(\mathcal{X}') - CQ_0(\mathcal{X})) \right)^2 \right]}$

***Shared LCB:*** $\mathrm{LCB}\left( Q_{t+1}(\mathcal{X}') \right) \approx (1 + \ldots + \gamma^t C^t) CR - (1 + \ldots + \gamma^t C^t)\sqrt{\mathbb{E}\left[ \left( Q_0(\mathcal{X}') - CQ_0(\mathcal{X}) \right)^2 \right]}$

*where LCB refers to mean - std, and the square and square-root operations are applied element-wise.*

*Proof.* Please refer to Appendix F. □

As can be seen, the equations for the lower-confidence bound (LCB) in both settings are very similar, with the main difference being in the second terms which correspond to the "pessimism" terms. In ensembles, the only source of randomness is in the initialization of the networks. In the infinite-width setting this presents itself in the two equations above, where the random variables $Q_0(\mathcal{X}') - CQ_0(\mathcal{X})$ produce the uncertainty in the ensemble of networks: For both Independent and Shared-LCB, at iteration $t + 1$ we have,

$$Q_{t+1}(\mathcal{X}') = Q_0(\mathcal{X}') + C(\mathcal{Y}_t - Q_0(\mathcal{X})) \tag{5}$$

$$= C\mathcal{Y}_t + (Q_0(\mathcal{X}') - CQ_0(\mathcal{X})) \tag{6}$$

Thus, $Q_0(\mathcal{X}') - CQ_0(\mathcal{X})$ represents the random value accumulated in each iteration. The accumulation of the uncertainty is captured by the geometric term $(1 + \ldots + \gamma^t C^t)$. Here is where we observe the key difference between Independent and Shared-LCB: whether the term $(1 + \ldots + \gamma^t C^t)$ is applied inside or outside the expectation. In Independent ensembles, the randomness/uncertainty is first backed-up by the geometric term and afterward the standard-deviation is computed. In Shared-LCB however, first the standard-deviation of the randomness/uncertainty is computed, and afterwards this value is backed up. Not only do we believe that the former (Independent) makes more sense intuitively, but in the case of Shared-LCB, *the pessimism term may contain negative values which would actually result in an optimism bonus*! As will be discussed below, we also empirically investigate this question in section 5.3.2 (results in Figure 2) and find that while Shared-LCB can perform decently in D4RL gym locmotion (Fu et al., 2020), Shared-LCB (and Shared-Min) completely fail on the more challenging domains. This is in line with the observation that no prior offline RL methods rely solely on Shared-LCB or Shared-Min as the source of pessimism/conservatism (Fujimoto et al., 2019; Kumar et al., 2019; Ghasemipour et al., 2021; An et al., 2021).

## 4.2 VALIDATING THEORETICAL PREDICTIONS

In this section we aim to evaluate the validity of our theoretical discussion above in a simple toy MDP that allows us to follow the idealized setting of the presented theorems more closely, and allows for visualization of uncertainties obtained through different ensembling approaches.

**Continuous Chain MDP**  The MDP we consider has state-space $\mathcal{S} = [-1, 1]$, action space $\mathcal{A} \in \mathbb{R}$, deterministic transition dynamics $s' = s + a$ clipped to remain inside $\mathcal{S}$, and reward function $r(s, a) = \mathbb{1}[s' \in [0.75, 1]]$.

**Data Collection & Evaluation Policy**  The offline dataset we generate consists of 40 episodes, each of length 30. At the beginning of each episode we initialize at a random state $s \in \mathcal{S}$. In each step take a random action sampled from $\text{Unif}(-0.3, 0.3)$, and record all transitions $(s, a, r, s')$. For evaluating the uncertainties obtained from different approaches, we create regions of missing data by removing all transitions such that $s$ or $s'$ are in the range $[-0.33, 0.33]$. The policy we choose to evaluate with the different approaches is $\forall s, \pi(s) = 0.1$.

**Optimal Desired Form of Uncertainty**  Note that the evaluation policy $\pi(s) = 0.1$ is always moving towards the positive direction, and there is lack of data for states in the interval $[-0.33, 0.33]$. Hence, what we would expect is that in the region $[0.33, 1]$ there should not be a significant amount of uncertainty, while in the region $[-1, -0.33]$ there should be significantly more uncertainty about the Q-values of $\pi$ because the policy will be passing through $[-0.33, 0.33]$ where there is no data.

**Results**  We visualize and compare the uncertainties obtained when the targets in the policy evaluation procedure are computed as:

- **Independent (MSG):** $y^i = r + \gamma \cdot Q_{\theta^i}(s', \pi(s'))$

- **Independent Double-Q:** $y^i = r + \gamma \cdot \min\left[Q^1_{\theta^i}(s', \pi(s')), Q^2_{\theta^i}(s', \pi(s'))\right]$

- **Shared Mean:** $y = r + \gamma \cdot \text{mean}\left[Q_{\theta^i}(s', \pi(s'))\right]$

- **Shared LCB:** $y = r + \gamma \cdot \left[\text{mean}\left[Q_{\theta^i}(s', \pi(s'))\right] - 2 \cdot \text{std}\left[Q_{\theta^i}(s', \pi(s'))\right]\right]$

- **Shared Min:** $y = r + \gamma \cdot \min\left[Q_{\theta^i}(s', \pi(s'))\right]$

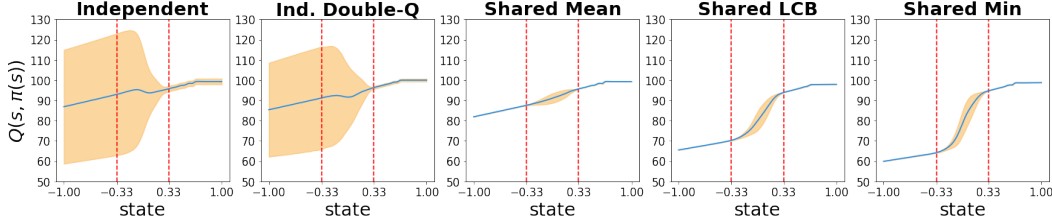

Figure 1: Verifying theoretical predictions on the toy Continuous Chain MDP. The marked interval $[-0.33, 0.33]$ denotes the region of state-space with no data. As anticipated by Theorem 4.1, when the value functions are trained independently, the derived uncertainties capture the interaction between available data, the structure of the MDP, and the policy being evaluated. When the targets are shared, the networks behave similarly to performing regression for oracle-given target values, i.e. there is randomness amongst ensemble members only between $[-0.33, 0.33]$ because there is no data in that region.

We include `Independent Double-Q`, as using Q-functions of the form $Q(s, a) = \min\left[Q^1(s, a), Q^2(s, a)\right]$ has become common practice in recent deep RL literature (Fujimoto et al., 2018; Haarnoja et al., 2018)[1].

In Figure 1 we plot the mean and two standard deviations of the predicted values for the policy we evaluated, $\pi(s) = 0.1$ (additional experimental details presented in Appendix G.1). The first thing to note is that, "Independent" ensembles effectively match our desired form of uncertainty: states that under the evaluation policy lead to regions with little data have wider uncertainties than states that do not. A second observation is that `Shared LCB` and `Shared Min` provide a seemingly good approximation to the lower-bound of `Independent` predictions. Nonetheless, our theoretical considerations suggest that these lower bounds may have important failure cases. Furthermore, empirically – as we discuss below (Figure 2) – we were unable to train effective policies on offline RL benchmarks using `Shared LCB` and `Shared Min`, despite the implementation differing in only 2 lines of code.

**Appendix G.2 presents additional very interesting empirical observations in this toy setting which due space limitations we were unable to include in the main manuscript.** We highly encourage readers interested in the intersection of infinite-width networks and RL to take a look at our observations as they may be hinting at intriguing avenues for future theoretical and practically important work.

## 5 EXPERIMENTS

In this section we seek to empirically answer the following questions: 1) How well does MSG perform compared to current state-of-the-art in offline RL? 2) Can we match the performance of MSG through efficient ensemble approaches popular in supervised learning literature? 3) Are the theoretical differences in ensembling approaches elaborated on in the previous section practically relevant?

### 5.1 D4RL BENCHMARK

We begin by evaluating MSG on the Gym (Brockman et al., 2016) subset of the D4RL offline RL benchmark (Fu et al., 2020). Amongst the different data settings we focus our experiments on the `medium` and `medium-replay` (sometimes referred to as `mixed`) settings as the other data setting could not adequately differentiate between competitive methods. Experimental details such as hyperparameter search procedure are described in Appendix C.

In addition to validating MSG, a secondary objective of ours is to gain a sense for the upper bound of performance for various algorithms on D4RL Gym. For this reason – with an equal hyperparameter tuning budget – we tune the main hyperparameter for each algorithm. As can be seen in Table 3 (Appendix A), across the board in D4RL Gym, MSG is competitive with very well-tuned current state-of-the-art algorithms, CQL (Kumar et al., 2020) and F-BRC (Kostrikov et al., 2021). We

---

[1]Note that `Independent Double-Q` is still an independent ensemble, where each ensemble member has an architecture containing a min-pooling on top of two subnetworks.

| Domain | CQL (Reported) | MSG ($N = 64$) | $\beta$ | $\alpha$ |
|---|---|---|---|---|
| maze2d-umaze | 5.7 | **100.2 ± 33.0** | -8 | 0 |
| maze2d-medium | 5.0 | **87.4 ± 10.6** | 0 | 0 |
| maze2d-large | 12.5 | **147.8 ± 55.8** | -4 | 0.1 |
| antmaze-umaze | 74.0 | **96.8 ± 2.0** | −8 | 0.1 |
| antmaze-umaze-diverse | **84.0** | 60.2 ± 7.1 | −8 | 0.5 |
| antmaze-medium-play | 61.2 | **80.0 ± 9.4** | −4 | 0.1 |
| antmaze-medium-diverse | 53.7 | **78.8 ± 5.5** | −4 | 0.1 |
| antmaze-large-play | 15.8 | **64.8 ± 10.7** | −8 | 0 |
| antmaze-large-diverse | 14.9 | **68.8 ± 11.7** | −8 | 0 |

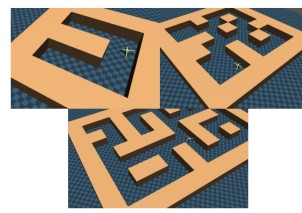

Table 2: D4RL antmaze tasks. Figure taken from Fu et al. (2020).

Table 1: Result on D4RL maze2d and antmaze domains. As we were unable to reproduce CQL antmaze results, we present the numbers reported by the original paper which uses the same network architectures.

also note that our results for baseline algorithms exceed prior reported results (often significantly), providing us confidence in their implementation.

Thus far we established the validity of MSG as an offline RL algorithm. However, on the gym domains considered above, MSG's performance is on par with CQL and F-BRC, and does not motivate the use of ensembles in lieu of support constraints. To enable a better comparison amongst competing algorithms, we experiment with the significantly more challenging antmaze tasks.

The antmaze tasks in D4RL, and in particular the two antmaze-large settings, are considered to be extremely challenging. The data for antmaze tasks consists of many episodes of an Ant agent (Brockman et al., 2016) running along arbitrary paths in a maze. The data from these trajectories is relabeled with a reward of 1 when near a particular location in the maze (at which point the episode is terminated), and 0 otherwise. The undirected, extremely sparse reward nature of antmaze tasks make them very challenging, especially for the large maze sizes.

*To the best of our knowledge, the antmaze-large domains are considered unsolved, unless specialized technique such as hierarchical policies – which significantly simplify the problem – are used (e.g. Ajay et al. (2020)).* The current state-of-the-art is CQL (Kumar et al., 2019), and is the method we compare to. Table 1 presents our empirical results. As we were unable to reproduce the reported results for CQL, for fairness we include the numbers reported by the original work which uses the same network architectures. As can be seen, MSG achieves unprecedented results on the antmaze domains, clearly demonstrating the significant advantage of employing ensembles for batch RL.

## 5.2 RL Unplugged

Figure 4 presents additional results using the RL Unplugged benchmark (Gulcehre et al., 2020). We compare to results for Behavioral Cloning (BC) and two state-of-the-art methods, Critic-Regularized Regression (CRR) (Wang et al., 2020) and MuZero Unplugged (Schrittwieser et al., 2021). Despite the relatively very small architectures we used ($\frac{1}{60}$ number of parameters), we observe that MSG is on par with the current state-of-the-art these tasks with the exception of humanoid.run which appears to require the larger architectures used by (Gulcehre et al., 2020). Additional experimental details as well as numerical presentation of results can be found in appendix E.

## 5.3 Efficient Ensembles & Ensemble Ablations

In this section we dissect what aspects of MSG contribute to its superior results, and investigate whether the advantages of MSG can be realized through efficient ensemble approximations popular in supervised learning.

### 5.3.1 Efficient Ensmebles

Thus far we have demonstrated the significant performance gains attainable through MSG. An important concern however, is that of parameter and computational efficiency: "Deep Ensembles" result in an $N$-fold increase in memory and compute usage for the Q-networks. While this might not be a significant problem for D4RL benchmark domains due to small model footprints , it becomes a major bottleneck with larger architectures such as those used in language and vision domains. To

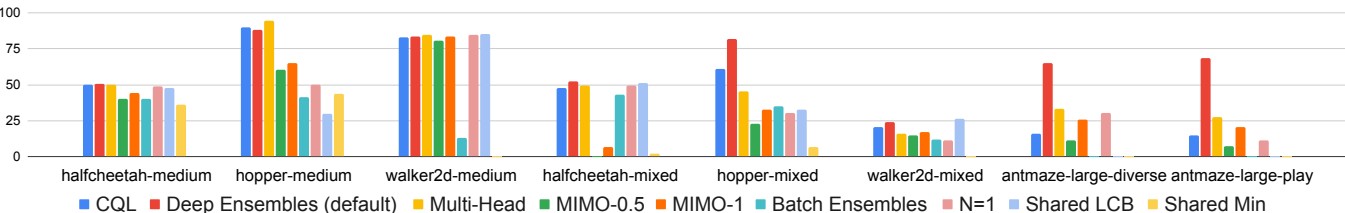

Figure 2: Results for efficient ensembles and ensemble ablations. Numerical values can be found in Table 4.

this end, we evaluate whether recent advances in efficient ensemble approaches also transfer well to the problem of batch RL. Specifically, the efficient ensemble approaches we consider are:

**Multi-Head (Lee et al., 2015; Osband et al., 2016; Tran et al., 2020)** Multi-Head refers to ensembles that share a "trunk" network and have separate "head" networks for each ensemble member. In this work, we modify the last layer of a value network to output $N$ predictions instead of a single Q-value, making the computational cost of this ensemble on par with a single network.

**Multi-Input Multi-Output (MIMO) (Havasi et al., 2020)** MIMO is an ensembling approach that approximately has the same parameter and computational footprint as a single network. The MIMO approach only modifies the input and output layers of a given network. In MIMO, to compute predictions for data-points $x_1, ..., x_N$ under an ensemble of size $N$, the data-points are concatenated and passes to the network. The output of the network is then split into $N$ predictions $y_1, ..., y_N$. For added clarification, we include Figure 3a depicting how a MIMO ensemble network functions.

**Batch Ensembles (Wen et al., 2020)** Batch Ensembles incorporate rank-1 modulations to the weights of fully-connected layers. More specifically, let $W$ be the weight matrix of a given fully-connected layer and let $x$ be the input to the layer. The output of the layer for ensemble member $i$ is computed as $\sigma(((W^T(x \circ r^i)) \circ s^i) + b^i)$, where $\circ$ is the element-wise product, parameters with superscript $i$ are separate for each ensemble member, and $\sigma$ is the activation function. While Batch Ensemble is efficient in terms of number of parameters, in our actor-critic setup its computational cost is on the same order as deep ensembles since for policy updates we need to evaluate each ensemble member separately.

**Results** As can be seen in Table 4, the performance gains of MSG can to some extent also be realized by efficient ensemble variants. Interestingly, in our experiments the most effective efficient ensemble approach is Multi-Head ensembles, which is not considered to be the most competitive uncertainty estimation technique in the supervised learning literature (Havasi et al., 2020). Compared to CQL, Multi-Head ensembles continue to be competitive on D4RL gym, and noticeably outperform CQL on antmaze-large tasks. Additionally, training Multi-Head ensembles is the most efficient method, on par with training without ensembling ($N = 1$).

Nonetheless, compared to MSG using deep ensembles, there is a significant performance gap. We believe this observation very clearly motivates future work in developing efficient uncertainty estimation approaches that are better suited to the domain of reinforcement learning. To facilitate this direction of research, in our codebase which will be open-sourced, we also include a complete boilerplate example amenable to drop-in implementation of novel uncertainty-estimation techniques.

### 5.3.2 Ensemble Ablations

Finally, through the ablations in Table 4 we seek to create a better sense for the various components of MSG.

- Comparing MSG with deep ensembles ($N = 64$), and Multi-Head ensembles ($N = 64$), to no ensembling ($N = 1$) we see very clearly the massive advantage of ensembling.

- Comparing MSG, which uses `Independent` targets, to `Shared LCB` and `Shared Min` we see that the latter non-independent approaches significantly underperform.

- Comparing CQL to $N = 1$ (no ensembling, only CQL-inspired regularizer), we observe that the CQL regularizer tends to be better on Gym domains, and our regularizer may be better on antmaze domains but our results are inconclusive. The advantage of the regularizer we used is significant computationl efficieny due to not using importance sampling.

## 6  DISCUSSION

Our work has highlighted the significant power of ensembling as a mechanism for uncertainty estimation for offline RL. Theoretically, and practically through benchmark experiments, we have studied the critical importance of the manner in which ensembling is done. An important outstanding direction is how can we design improved efficient ensemble approximations, as we have demonstrated that current approaches used in supervised learning – some of which do lead to state-of-the-art results offline RL results – are not nearly as effective as deep ensembles. We hope that this work engenders new efforts from the community of deep network uncertainty estimation researchers whom thus far have not employed offline reinforcement learning domains as a testbed for validating modern uncertainty estimation techniques.

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

## A  D4RL Gym Locomotion Benchmark Table

| Domain | BC | CQL | $\alpha$ | F-BRC (no bonus) | $\alpha$ | MSG ($N = 64$) | $\beta$ | $\alpha$ | Data Top 25 |
|---|---|---|---|---|---|---|---|---|---|
| halfcheetah-medium | 36.1 | $\mathbf{50.1 \pm 1.2}$ | 0 | $45.7 \pm 0.5$ | 0.01 | $\mathbf{50.6 \pm 1.5}$ | 0 | 0 | $37.0 \pm 2.0$ |
| hopper-medium | 29.0 | $90.2 \pm 10.5$ | 2.58 | $\mathbf{99.0 \pm 4.1}$ | 0.012 | $88.2 \pm 11.6$ | -1 | 0.5 | $101.1 \pm 1.1$ |
| walker2d-medium | 6.6 | $\mathbf{82.8 \pm 0.6}$ | 3.38 | $81.6 \pm 0.2$ | 0.016 | $83.6 \pm 2.0$ | -1 | 0.1 | $77.6 \pm 1.4$ |
| halfcheetah-mixed | 38.4 | $47.6 \pm 2.5$ | 0.67 | $45.3 \pm 0.1$ | 0.01 | $\mathbf{52.1 \pm 1.7}$ | 0 | 0 | $38.1 \pm 4.1$ |
| hopper-mixed | 11.8 | $61.1 \pm 36.8$ | 0.67 | $29.3 \pm 0.8$ | 0.019 | $\mathbf{82.1 \pm 24.2}$ | -2 | 0 | $101.8 \pm 1.5$ |
| walker2d-mixed | 11.3 | $\mathbf{20.5 \pm 3.8}$ | 3.38 | $\mathbf{28.4 \pm 5.7}$ | 0.038 | $24.0 \pm 9.4$ | -2 | 0.1 | $50.6 \pm 3.9$ |

Table 3: Results on Gym subset of the D4RL benchmark. For MSG we use an ensemble size of $N = 64$. For fairness of comparison, F-BRC is ran without adding a survival reward bonus. For each method we also report the hyperparameter value used. Full experimental details are presented in Appendix C.

## B  Efficient Ensembles & Ensemble Ablations Table

| Domain | CQL | Deep Ens. | MIMO-0.5 | MIMO-1 | Multi-Head | Batch Ens | $N = 1$ | Shared LCB | Shared Min |
|---|---|---|---|---|---|---|---|---|---|
| halfcheetah-medium | 50.1 | 50.6 | 40.0 | 44.4 | 49.8 | 40.2 | 48.6 | 47.9 | 36.0 |
| hopper-medium | 90.2 | 88.2 | 60.2 | 65.1 | 94.8 | 41.6 | 50.2 | 30.0 | 43.6 |
| walker2d-medium | 82.8 | 83.6 | 80.7 | 83.6 | 84.5 | 13.2 | 84.9 | 85.5 | 0.0 |
| halfcheetah-mixed | 47.6 | 52.1 | 0.0 | 6.7 | 49.4 | 43.0 | 49.6 | 51.0 | 2.3 |
| hopper-mixed | 61.1 | 82.1 | 22.7 | 32.5 | 45.5 | 35.0 | 30.5 | 32.5 | 6.7 |
| walker2d-mixed | 20.5 | 24.0 | 14.5 | 16.8 | 15.9 | 12.0 | 11.5 | 26.3 | 0.0 |
| antmaze-large-diverse | 15.8 | 64.8 | 11.0 | 26.0 | 33.5 | 0.0 | noisy 30.5 | 0.0 | 0.0 |
| antmaze-large-play | 14.9 | 68.8 | 7.5 | 20.5 | 27.5 | 0.0 | noisy 11.0 | 0.0 | 0.0 |

Table 4: Results for efficient ensembles and ensemble ablations.

## C  D4RL Gym Details

All policies and Q-functions are a 3 layer neural network with relu activations and hidden layer size 256. The policy output is a normal distribution that is squashed to $[-1, 1]$ using the tanh function. All methods were trained for 1M steps. CQL and MSG are trained with behavioral cloning (BC) for the first 50K steps. F-BRC pretrains with 1M steps of BC.

CQL, F-BRC, and MSG are tuned with an equal hyperparameter search budget. Running the best found hyperparameter using 5 new random seeds. Each run is evaluated for 10 episodes, and the mean and standard deviation across the 5 runs is reported in the table. For fairness of comparison, F-BRC is ran without adding a survival reward bonus. MSG and CQL are implemented in our code, and for F-BRC we use the opensourced codebase.

### C.1  Hyperparameter Search

For all methods we performed hyperparameter search using 2 seeds. Based on the results of the 2 seeds, we chose the hyperparameter to use, and using this choice of hyperparameter we ran experiments with 5 new random seed.

**MSG**  $\beta \in \{0., -1., -2., -4. - 8.\}, \alpha \in \{0., 0.1, 0.5, 1., 2.\}$

**CQL**  $\alpha \in \{0., 0.1\} + np.exp(np.linspace(np.log(0.1), np.log(10.), steps = 23))$

**F-BRC**  $\alpha \in \{0.\} + np.exp(np.linspace(np.log(0.01), np.log(10.0), steps = 24))$

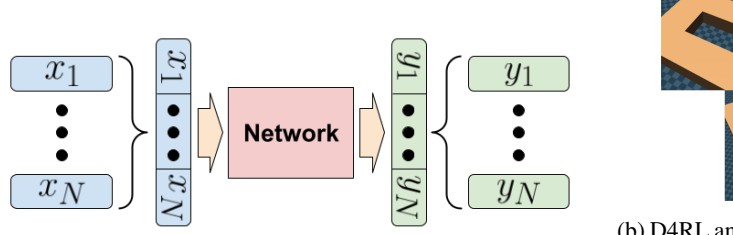

(a) Visual depiction of MIMO Ensemble

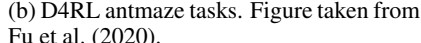

(b) D4RL antmaze tasks. Figure taken from Fu et al. (2020).

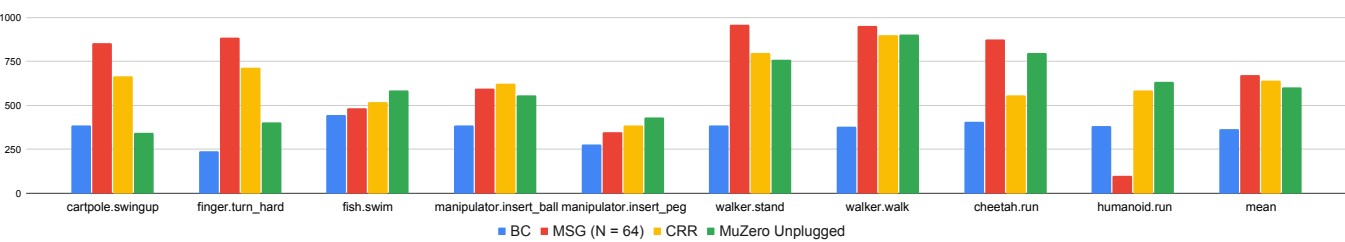

Figure 4: Results for DM Control Suite subset of the RL Unplugged benchmark (Gulcehre et al., 2020). We note that: 1) the architecture we used are smaller by a factor of approximately 60x, 2) CRR results are reported by their best checkpoint throughout training which differs from ours, BC, and MuZero Unplugged which report performance at the end of training. Baseline results taken from Schrittwieser et al. (2021).

## D    Antmaze Details

We use the same hyperparameter search procedure as Gym results, with same architectures. The only difference is that models are trained for 2M steps and at evaluation time they are rolled out for 100 episodes instead of 10.

In prior work, the rewards in the offline dataset are converted using the formula $4(r - 0.5)$. We also use the same reward transformation.

## E    RL Unplugged

### E.1    DM Control Suite Tasks

The networks used in Gulcehre et al. (2020) for DM Control Suite Tasks are very large relative to the networks we used in the D4RL benchmark; roughly the networks contain 60x more parameters. Using a large ensemble size with such architectures requires training using a large number of devices. Furthermore, since in our experiments with efficient ensemble approximations we did not find a suitable alternative to deep ensembles (section 5.3.1), we decided to use the same network architectures and $N = 64$ as in the D4RL setting (enabling single-GPU training as before).

Our hyperparameter search procedure was similar to before, where we first performed a coarse search using 2 random seeds and hyperparameters $\beta \in \{-1., -2., -4. -8.\}, \alpha \in \{0., 0.5\}$, and for the best found hyperparameter, ran final experiments with 5 new random seeds.

## F    Theory

For our notation to match Lee et al. (2019), throughout this section we use the $f$ to denote the network $Q$, and we use $x$ instead of $(s, a)$.

In Lee et al. (2019) (section 2.4) it is shown that when training an infinitely wide neural network to perform regression using mean squared error, subject to technical conditions on the learning rate used, the predictions of the trained network are equivalent to if we had linearized (Taylor expanded)

the network at its initialization, and trained the linearized network instead. This means that after $t$ iterations of our policy evaluation procedure, $\forall x, t, f_t^{\text{lin}}(x) = f_t(x)$, where

$$f_t^{\text{lin}}(x) := f_0(x) + \nabla_\theta f_0(x)|_{\theta=\theta_0} \, \omega_t$$
$$\omega_t := \theta_t - \theta_0$$

Hence we only need to study the evolution of the linearized network $f_t^{\text{lin}}$ across iterations. The theorems in the main manuscript are direct corollaries of the following two derivations.

Below, we will overload some notation. When $f^{\text{lin}}$ is applied to a matrix we mean that we apply $f^{\text{lin}}$ to each row of the matrix and stack the results into a vector. By $\hat{\Theta}_0(x, \mathcal{X})$ we mean to treat $x$ as a row matrix.

### F.1 CLOSED FORM WHEN ENSEMBLE MEMBERS USE THEIR OWN TARGETS

For a single – infinitely wide – ensemble member, using the equations in Lee et al. (2019) (section 2.2, equations 9-10-11) we can write the following recursive updates for our policy evaluation procedure

$$\hat{\Theta}_0^{-1} := \hat{\Theta}_0(\mathcal{X}, \mathcal{X})^{-1} \tag{7}$$

$$C := \hat{\Theta}_0(\mathcal{X}', \mathcal{X})\hat{\Theta}_0^{-1} \tag{8}$$

$$\mathcal{Y}_t = R + \gamma f_t^{\text{lin}}(\mathcal{X}') \tag{9}$$

$$f_{t+1}^{\text{lin}}(\mathcal{X}) = \mathcal{Y}_t \tag{10}$$

$$\forall x, f_{t+1}^{\text{lin}}(x) = f_0(x) + \hat{\Theta}_0(x, \mathcal{X})\hat{\Theta}_0^{-1}(\mathcal{Y}_t - f_0(\mathcal{X})) \tag{11}$$

$$f_{t+1}^{\text{lin}}(\mathcal{X}') = f_0(\mathcal{X}') + \hat{\Theta}_0(\mathcal{X}', \mathcal{X})\hat{\Theta}_0^{-1}(\mathcal{Y}_t - f_0(\mathcal{X})) \tag{12}$$

$$= f_0(\mathcal{X}') + \hat{\Theta}_0(\mathcal{X}', \mathcal{X})\hat{\Theta}_0^{-1}(R + \gamma f_t^{\text{lin}}(\mathcal{X}') - f_0(\mathcal{X})) \tag{13}$$

$$= f_0(\mathcal{X}') + CR + \gamma C f_t^{\text{lin}}(\mathcal{X}') - C f_0(\mathcal{X}) \tag{14}$$

$$= f_0(\mathcal{X}') + CR - C f_0(\mathcal{X}) + \gamma C f_t^{\text{lin}}(\mathcal{X}') \tag{15}$$

$$= \ldots \tag{16}$$

$$= (1 + \ldots + \gamma^t C^t)\Big(f_0(\mathcal{X}') + CR - C f_0(\mathcal{X})\Big) + (\gamma C)^{t+1} f_0(\mathcal{X}') \tag{17}$$

$$\mathbb{E}[f_{t+1}^{\text{lin}}(\mathcal{X}')] = (1 + \ldots + \gamma^t C^t)CR \tag{18}$$

$$\text{Var}[f_{t+1}^{\text{lin}}(\mathcal{X}')] = \mathbb{E}\Big[\Big((1 + \ldots + \gamma^t C^t)(f_0(\mathcal{X}') - C f_0(\mathcal{X})) + (\gamma C)^{t+1} f_0(\mathcal{X}')\Big)^2\Big] \tag{19}$$

$$\approx \mathbb{E}\Big[\Big((1 + \ldots + \gamma^t C^t)(f_0(\mathcal{X}') - C f_0(\mathcal{X}))\Big)^2\Big] \quad \text{as } t \to \infty \tag{20}$$

### F.2 CLOSED FORM WHEN ENSEMBLE MEMBERS USE SHARED MEAN TARGETS

Let us consider the setting where all ensemble memebers use their mean as the target.

$$\hat{\Theta}_0^{-1} := \hat{\Theta}_0(\mathcal{X}, \mathcal{X})^{-1} \tag{21}$$

$$C := \hat{\Theta}_0(\mathcal{X}', \mathcal{X})\hat{\Theta}_0^{-1} \tag{22}$$

$$\mathcal{Y}_t = R + \gamma f_t^{\text{lin}}(\mathcal{X}') \tag{23}$$

$$f_{t+1}^{\text{lin}}(\mathcal{X}) = \mathcal{Y}_t \tag{24}$$

$$\forall x, f_{t+1}^{\text{lin}}(x) = f_0(x) + \hat{\Theta}_0(x, \mathcal{X})\hat{\Theta}_0^{-1}(\mathcal{Y}_t - f_0(\mathcal{X})) \tag{25}$$

$$\mathcal{Y}_t = R + \gamma \mathop{\mathbb{E}}_{ensemble}[f_t^{\text{lin}}(\mathcal{X}')] \tag{26}$$

$$= R + \gamma \mathop{\mathbb{E}}_{ensemble}[f_0(\mathcal{X}')] + \gamma C\mathcal{Y}_{t-1} - \gamma C \mathop{\mathbb{E}}_{ensemble}[f_0(\mathcal{X})] \tag{27}$$

$$= R + \gamma C\mathcal{Y}_t \tag{28}$$

$$= \ldots \tag{29}$$

$$= (1 + \ldots + \gamma^t C^t)R \tag{30}$$

### F.3    Closed form when ensemble members use shared LCB targets

Let us consider the setting where all ensemble memebers use shared LCB as the target.

$$\hat{\Theta}_0^{-1} := \hat{\Theta}_0(\mathcal{X}, \mathcal{X})^{-1} \tag{31}$$

$$C := \hat{\Theta}_0(\mathcal{X}', \mathcal{X})\hat{\Theta}_0^{-1} \tag{32}$$

$$\mathcal{Y}_t = \text{LCB}\left(R + \gamma f_t^{\text{lin}}(\mathcal{X}')\right) \tag{33}$$

$$= R + \gamma\text{LCB}\left(f_t^{\text{lin}}(\mathcal{X}')\right) \tag{34}$$

$$f_{t+1}^{\text{lin}}(\mathcal{X}) = \mathcal{Y}_t \tag{35}$$

$$\forall x, f_{t+1}^{\text{lin}}(x) = f_0(x) + \hat{\Theta}_0(x, \mathcal{X})\hat{\Theta}_0^{-1}(\mathcal{Y}_t - f_0(\mathcal{X})) \tag{36}$$

$$\mathbb{E}[f_{t+1}(\mathcal{X}')] = \mathbb{E}[f_0(\mathcal{X}') + C \cdot (\mathcal{Y}_t - f_0(\mathcal{X}))] \tag{37}$$

$$= C \cdot \mathcal{Y}_t \tag{38}$$

$$\text{Var}[f_{t+1}(\mathcal{X}')] = \mathbb{E}\left[\left(f_0(\mathcal{X}') - C \cdot f_0(\mathcal{X})\right)^2\right] = \text{constant} \tag{39}$$

$$A := \sqrt{\text{constant}} \tag{40}$$

$$\text{LCB}\left(f_{t+1}^{\text{lin}}(\mathcal{X}')\right) = C \cdot \mathcal{Y}_t - A \tag{41}$$

$$\mathcal{Y}_t = R + \gamma\text{LCB}\left(f_t^{\text{lin}}(\mathcal{X}')\right) \tag{42}$$

$$= R + \gamma C\mathcal{Y}_t - \gamma A \tag{43}$$

$$= (1 + \ldots + \gamma^t C^t)R - (1 + \ldots + \gamma^{t-1}C^{t-1})\gamma A + \gamma^{t+1}C^t\text{LCB}\left(f_0(\mathcal{X}')\right) \tag{44}$$

$$\approx (1 + \ldots + \gamma^t C^t)R - (1 + \ldots + \gamma^{t-1}C^{t-1})\gamma A \quad \text{as } t \to \infty \tag{45}$$

$$\text{LCB}\left(f_{t+1}^{\text{lin}}(\mathcal{X}')\right) \approx (1 + \ldots + \gamma^t C^t)CR - (1 + \ldots + \gamma^t C^t)A \tag{46}$$

### F.4    Why is Independent preferable to Shared-LCB

An important question to consider is why Independent ensembles should be preferred over Shared-LCB ensembles? Here we present our reasoning for why we Independent ensembles would be preferable to Shared-LCB ensembles.

With the derivations in the above sections, we can compare the difference amongst uncertainty estimation techniques. The key comparison needed is to understand the difference between

$\text{LCB}\left(f_{t+1}^{\text{lin}}(\mathcal{X}')\right)$ under Independent vs. Shared-LCB settings. As a reminder, $\mathcal{X}'$ is a matrix where each row contains $(s, \pi(s))$, and $f_{t+1} = f_{t+1}^{\text{lin}}$ under the infinite-width regime. From the above equations we have:

**Independent:** $\text{LCB}\left(f_{t+1}^{\text{lin}}(\mathcal{X}')\right) \approx (1 + \ldots + \gamma^t C^t)CR - \sqrt{\mathbb{E}\left[\left((1 + \ldots + \gamma^t C^t)(f_0(\mathcal{X}') - Cf_0(\mathcal{X}))\right)^2\right]}$

**Shared-LCB:** $\text{LCB}\left(f_{t+1}^{\text{lin}}(\mathcal{X}')\right) \approx (1 + \ldots + \gamma^t C^t)CR - (1 + \ldots + \gamma^t C^t)\sqrt{\mathbb{E}\left[\left(f_0(\mathcal{X}') - Cf_0(\mathcal{X})\right)^2\right]}$

where the square and square-root operations are applied element-wise to the vector values.

As can be seen, the equations for the lower-confidence bound (LCB) in both settings are very similar, with the main difference being in the second terms which correspond to the "pessimism" terms. In the infinite-width setting, the only source of randomness is in the initialization of the networks. This fact presents itself in the two equations above, where the random variables $f_0(\mathcal{X}') - Cf_0(\mathcal{X})$ produce the uncertainty in the ensemble of networks; regardless of using Independent or Shared-LCB, after any iteration $t$ we have,

$$\mathcal{Y}_t = R + \gamma f_t(\mathcal{X}') \tag{47}$$
$$f_{t+1}(\mathcal{X}') = f_0(\mathcal{X}') + C(\mathcal{Y}_t - f_0(\mathcal{X})) \tag{48}$$
$$= C\mathcal{Y}_t + (f_0(\mathcal{X}') - Cf_0(\mathcal{X})) \tag{49}$$
$$\mathbb{E}[f_{t+1}(\mathcal{X}')] = C\mathbb{E}[\mathcal{Y}_t] \tag{50}$$

Thus, $f_0(\mathcal{X}') - Cf_0(\mathcal{X})$ represents the random value accumulated in each iteration, and they are accumulated through backups by the geometric term $(1 + \ldots + \gamma^t C^t)$.

Here is where we observe the key difference between Independent and Shared-LCB: whether the term $(1 + \ldots + \gamma^t C^t)$ is applied inside or outside the expectation. In Independent ensembles, the randomness/uncertainties is first backed-up by the geometric term and afterward the standard-deviation is computed. In Shared-LCB however, first the standard-deviation of the randomness/uncertainties is computed, and afterwards this value is backed up. Not only do we believe that the former (Independent) makes more sense intuitively, but in the case of Shared-LCB, the second term may contain *negative values* which would actually result in an *optimism bonus*!

### F.5 COMPARING THE STRUCTURE OF UNCERTAINTIES UNDER INDEPENDENT AND SHARED-MEAN

The above results enable us to compute in closed form the predictions of the infinite-width networks under different training regimes $\forall x$.

The equations above present the closed form expressions for the predictions of each ensemble member after $t+1$ iterations of the policy evaluation procedure. Since the ensemble members only differ in their weight initialization (random draws from the initial weight distribution), the random variables are $f_0(x), f_0(\mathcal{X}), f_0(\mathcal{X}')$. As mentioned in the main text, the neural tangent $\hat{\Theta}_0$ is identical across ensemble members due to being in the infinite-width regime (Jacot et al., 2018).

Since $\forall x, \underset{ensemble}{\mathbb{E}}[f_0(x)] = 0$ (Lee et al., 2019; 2017; Matthews et al., 2018), **the expected values of $f_{t+1}(x)$ is identical in for both methods of computing TD targets,**

$$\underset{ensemble}{\mathbb{E}}[\mathcal{Y}_{t+1}] = (1 + \ldots + \gamma^t C^t)R \tag{51}$$
$$\forall x, \underset{ensemble}{\mathbb{E}}[f_{t+1}(x)] = \hat{\Theta}_0(x, \mathcal{X})\hat{\Theta}_0^{-1}(1 + \ldots + \gamma^t C^t)R \tag{52}$$

**However, the expression for variances is very different.** When the targets used are independent we have,

$$\forall x, \underset{ensemble}{\text{Var}}[f_{t+1}(x)]$$
$$= \underset{ensemble}{\mathbb{E}}\left[\left(f_0(x) + \hat{\Theta}_0(x, \mathcal{X})\hat{\Theta}_0^{-1}\left((1 + \ldots + \gamma^t C^t)\left(\gamma f_0(\mathcal{X}') - f_0(\mathcal{X})\right)\right)\right)^2\right] \tag{53}$$

In contrast, when the targets are shared mean of targets, we have,

$$\forall x, \operatorname*{Var}_{ensemble}[f_{t+1}(x)] = \mathop{\mathbb{E}}_{ensemble} \left[ \left( f_0(x) + \hat{\Theta}_0(x, \mathcal{X})\hat{\Theta}_0^{-1}\left( - f_0(\mathcal{X})\right)\right)^2 \right] \tag{54}$$

The matrix $C$ captures a notion of similarity between the $(s, a)$ in $\mathcal{X}$, and the $(s', \pi(s'))$ in $\mathcal{X}'$. Thus, the term $C^t$ has the interpretation of where the policy $\pi(s)$ would find itself $t$ steps into the future, and $(1 + \ldots + \gamma^t C^t)$ can be interpreted as the policy's discounted state-action visitation, but in the feature-space given by the neural network architecture. Since in ensembles the standard deviation of predictions quantifies the amount of uncertainty, **the expression in equation 53 tells us that when the targets are independent, the ensemble of Q-functions "backs up the uncertainties through dynamic programming with respect to the policy being evaluated".**

**In contrast, when the targets are shared, the closed form expression for $f_{t+1}(x)$ is equivalent to an oracle presenting us with targets $(1 + \ldots + \gamma^t C^t)R$ for training examples $\mathcal{X}$, and training the ensemble members using mean squared error regression to regress these values.**

## G  ADDITIONAL TOY EXPERIMENTS & DETAILS

### G.1  ADDITIONAL IMPLEMENTATION DETAILS FOR FIGURE 1

To evaluate the quality of uncertainties obtained from different Q-function ensembling approaches, we create $N = 64$ Q-function networks, each being a one hidden layer neural network with hidden dimension 512 and `tanh` activation. The initial weight distribution is a fan-in truncated normal distribution with scale 10.0, and the initial bias distribution is fan-in truncated normal distribution with scale 0.05. We did not find results with other activation functions and choices of initial weight and bias distribution to be qualitatively different. We use discount $\gamma = 0.99$ and the networks are optimized using the Adam (Kingma & Ba, 2014) optimizer with learning rate `1e-4`. In each iteration, we first compute the TD targets using the desired approach (e.g. independent vs. shared targets) and then fit the Q-functions to their respective targets with 2000 steps of full batch gradient descent. We train the networks for 1000 such iterations (for a total of $2000 \times 1000$ gradient steps). Note that we do not use target networks. Given the small size of networks and data, these experiments can be done within a few minutes using a single GPU in `Google Colaboratory` which we will also opensource.

### G.2  ADDITIONAL TOY EXPERIMENTS

The toy experiment presented in section 4.2 uses a single-hidden layer finite-width neural network architecture with `tanh` activations, uses the "standard weight parameterization" (i.e. the weight parameterization used in practice) as opposed to the NTK parameterization (Novak et al., 2019), and optimizes the networks using the Adam optimizer (Kingma & Ba, 2014). While this setup is close to the practical setting and demonstrates the relevance of our proposal for independent ensembles for the practical setting, an important question posed by our reviewers is how close these results are too the theoretical predictions present in 4.1. To answer this question, we present the following set of results.

Using the identical MDP and offline data as before, we implement 1 hidden layer neural networks with `erf` non-lineartiy. The networks are implemented using the Neural Tangents library (Novak et al., 2019), and use the NTK parameterization. The networks in the ensemble are optimized using full-batch gradient descent with learning rate 1 for 500 steps of FQE Fonteneau et al. (2013), where each in each step the networks are updated for 1000 gradient steps. We vary the width of the networks from 32 to 32768 in increments of a factor of 4, plotting the mean and standard deviation of the network predictions. The ensemble size is set to $N = 16$, except for width 32768 where $N = 4$.

We compare the results from finite-width networks to computing the mean and standard deviation in closed for using 4.1. Using the Neural Tangents library (Novak et al., 2019) we obtained the NTK for the architecture described in the previous paragraph (1 hidden layer with `erf` non-linearity). We found that the matrix inversion required in our equations results in numerical errors. Hence, we make the modification $\Theta(\mathcal{X}, \mathcal{X}) \leftarrow \Theta(\mathcal{X}, \mathcal{X}) + \texttt{1e-3} \cdot \mathrm{I}$.

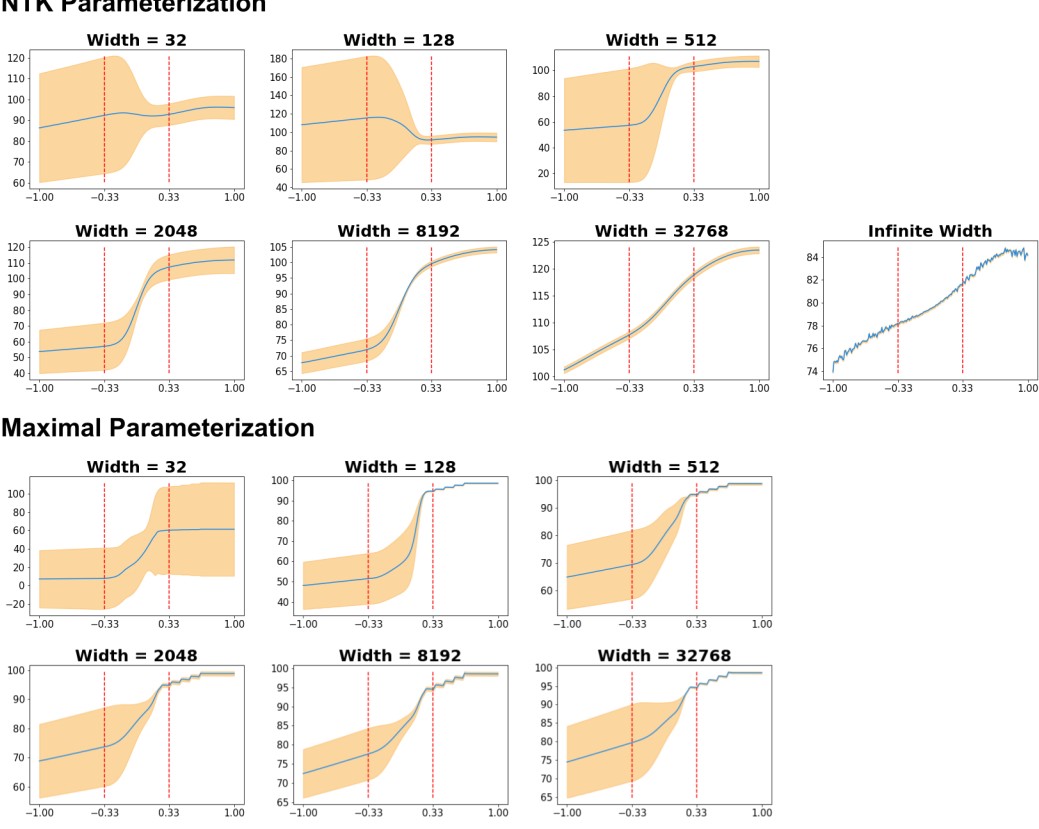

Figure 5: Comparing results of finite-width networks to closed form equations derived in Theorem 4.1. In the NTK parameterization, as width $\rightarrow \infty$, the structure of the variances collapse and resemble the infinite-width closed-form results. We believe this is due to infinite-width networks under the NTK regime not being able to learn features (Yang & Hu, 2020). Supporting this hypothesis, we observe that networks parameterized by the Maximal Parameterization of Yang & Hu (2020) maintain the desired uncertainty structure as the width of the networks grows larger.

Figure 5 presents our results. As the width of the networks grow larger, the shape of the uncertainties becomes more similar to our closed-form equations (i.e. the variances become very small). While we do not have a rigorous explanation for why finite-width networks exhibit intuitively more desirable behaviors, we present below a strong hypothesis backed by empirical evidence. We believe rigorously answering this question is an incredibly interesting avenue for future work.

Hypothesis: Infinite-width networks in the NTK parameterization/regime do not learn data-dependent features (Yang & Hu, 2020). Furthermore, as can be seen in the equations of Theorem 4.1, the variances depend the function values and features (kernel, $C$ matrix, etc.) *at initialization*. Yang & Hu (2020) present a different approach for parameterizing infinite-width networks called the "Maximal Parameterization", which enables inifinite-width networks to learn data-dependent features. We perform the same experiment as above, by replacing the NTK-parameterized networks with Maximal Parameterizations. Figure 5 presents our empirical results for network widths from 32 to 32768. **Excitingly, we observe that with Maximial Parameterization, even our widest networks recover the intuitively desired form of uncertainty described in section 4.2!** The solutions of these networks also appear much more accurate, particularly on the right hand side of the plot where with the stepped structure; each step appears to be approximately 0.1 in width, which is the action of the policy being evaluated.

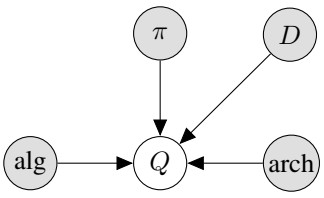

(a) Q-function generative process: the graphical model above represents the induced distribution over Q-functions when conditioning on a particular policy evaluation algorithm, policy, offline dataset, and Q-function network architecture.

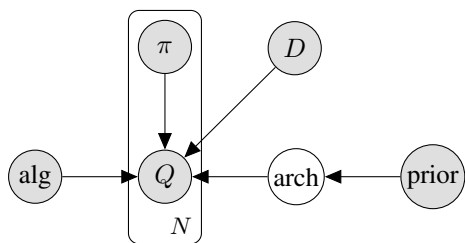

(b) An example of an interesting extension

# H    STATISTICAL MODEL

An interesting question posed by reviewers of our work was "[W]hatever formal reasoning system we'd like to use, what is the ideal answer, given access to arbitrary computational resources, so that approximations are unnecessary? I.e., how do we quantify our uncertainty about the MDP and value function before seeing data, and how do we quantify it after seeing data?"

It is important to begin by clarifying what is the mathematical object we are trying to obtain uncertainties over. In this work, we do not quantify uncertainties about any aspects of the MDP itself (although this is an interesting question which comes up in model-based methods as well as other settings such as Meta-Learning (Yu et al., 2019; Ghasemipour et al., 2019)). Our goal in this work is to directly estimate $Q^\pi(s, a) = r(s, a) + \gamma \cdot \mathbb{E}_{s' \sim MDP, a' \sim \pi}[Q(s', a')]$, for $a \sim \pi(s)$, and obtain uncertainties about $Q^\pi(s, a)$.

Let $Q(s, a)$ be a predictor $\mathcal{S} \times \mathcal{A} \to \mathbb{R}$ that needs to be evaluated on – and hopefully generalize well to – $(s, a) \notin D$ ($D$ being the offline dataset). When we choose to represent $Q(s, a)$ using neural networks, Gaussian Processes, or K-nearest-neighbours, we are not just making approximations for computational reasons, but are actually choosing a function class which we believe will generalize well to unseen $(s, a)$.

One practical example of learning Q-functions is to use Fitted Q iteration (Fonteneau et al., 2013) on the provided data using gradient descent with a particular neural network architecutre. Due to the random weight initialization, this procedure induces a distribution on the Q-functions which is captured by the probabilistic graphical model (PGM) in Figure 6a. In other words, by conditionining on the policy, data, architecture, and policy evaluation algorithm, we are imposing a belief over Q-functions. Note that this is essentially the same justification as using ensembles in supervised deep learning, where ensembles are state-of-the-art for accuracy and calibration (Ovadia et al., 2019). For the sake of theoretical analysis (Section 4), we studied this belief distribution under the infinite-width NTK network setting, in which case the distribution over Q-functions is a Gaussian Process.

The focus of this work is to ask the question: "Under this imposed belief, what should the policy update be?". Our proposed answer is to optimize the policy with respect to the lower-confidence bound of our beliefs: In an actor-critic setup, the policy optimization objective takes a form like $\max_\pi \mathbb{E}_{d(s)}[Q(s, \pi(s))]$, where $d(s)$ is some distribution over states (e.g. initial state distribution, or the states in the offline dataset $D$, etc.). Thus, our proposed policy objective takes the form $\max_\pi \text{LCB}\Big(\mathbb{E}_{d(s)}[Q(s, \pi(s))]\Big)$, and for practical reasons, in MSG we convert this to $\max_\pi \mathbb{E}_{d(s)}\Big[\text{LCB}\Big(Q(s, \pi(s))\Big)\Big]$ (which is a lower-bound of the first).

The graphical model in Figure 6a also highlights an example of interesting future directions: Consider an offline RL setup where we keep track of the various policies generating the data, and their Q-functions. Then, by imposing a prior on the architecture we can first infer a posterior distribution over architectures, then learn the Q-function of a new policy under the posterior architecture distribution.

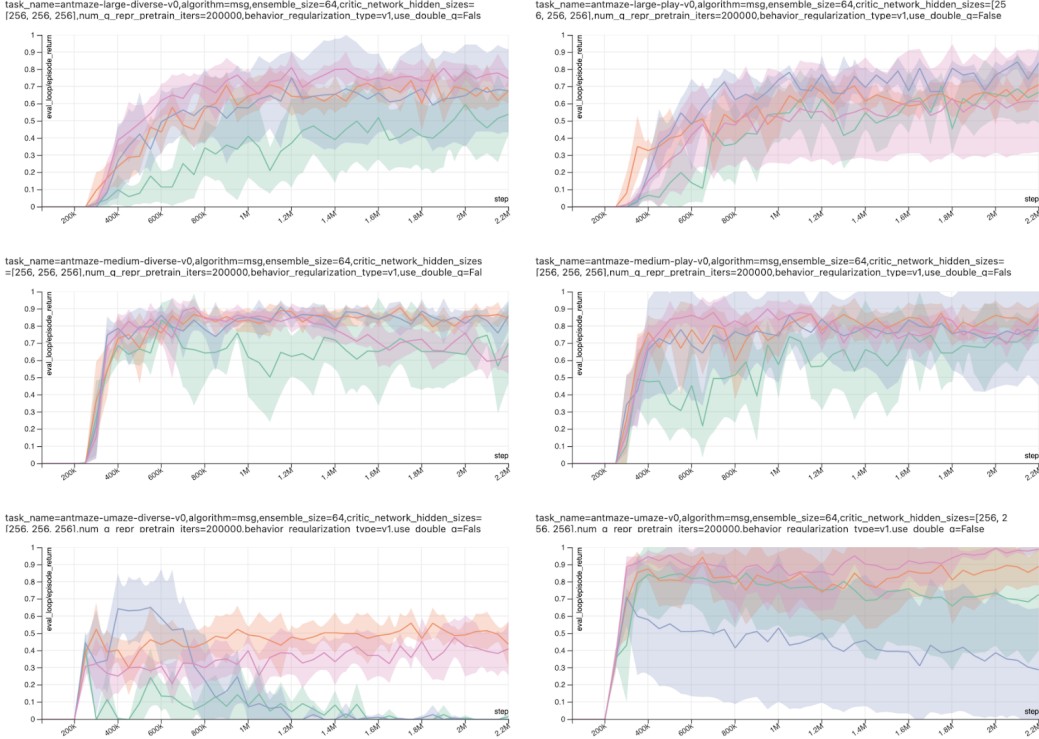

Figure 7: Results on the six antmaze domains. Each color represents the mean and standard deviation of results across 5 seeds for a particular hyperparameter setting ($\beta \in -4, -8$ and $\alpha \in 0, 0.1$). As can be seen, our results are quite robust across a wide range of hyperparameter values.

## I  PRACTICAL HYPERPARAMETER TUNING ADVICE

In reporting our results, we preferred to also report the hyperparameter values that we used, which may given an impression of significant hyperparameter tuning. We emphasize that our results are generally robust across a range of hyperparameter values. As an example, in Figure 7 we present results on the six antmaze domains. Each color represents the mean and standard deviation of results across 5 seeds for a particular hyperparameter setting ($\beta \in -4, -8$ and $\alpha \in 0, 0.1$). As can be seen, our results are quite robust across a wide range of hyperparameter values.

Here, we include practical advice on hyperparameter tuning when faced with a new domain. For a given new domain, we would first use low $\beta$ values: $\beta \in \{-4, -8\}$. If $\beta = -4$ is clearly better than $\beta = -8$, then we would guess that in this new domain high pessimism may not be necessary and explore the use of $\beta \in \{0, -1, -2\}$. For the $\alpha$ hyperparameter, we found that $\{0, 0.1, 0.5, 1.0\}$ is a wide enough range to explore. If $\alpha = 1.0$ was clearly better than the other values, then this would indicate to us that maybe the offline dataset is narrow (lacks diversity, e.g. imitation learning datasets) and then we could increase the value of $\alpha$. Generally, we would prefer to lower $\beta$ before attempting to increase $\alpha$, but we never tried $\beta < -8$.

