# OpenReview forum: "Why so pessimistic? Estimating uncertainties for offline RL through ensembles, and why their independence matters."
_ICLR.cc/2022/Conference — ICLR 2022 Submitted_

### Official Review · Reviewer_s2kZ · 2021-10-30

**Correctness:** 3
**Technical Novelty And Significance:** 2
**Empirical Novelty And Significance:** 2
**Recommendation:** 5
**Confidence:** 4

**Main Review:**

Strength:
The work proposes a novel method to use independent ensembles, which might be interesting towards a better understanding of uncertainty quantification in offline RL.
The experiment results are encouraging, showing superior performance of the proposed methods.
The theoretical intuitions are backed up with simulations, which helps readers better understand the ideas.

Weakness: I have several concerns that are listed below.
Intuitively, the method of using independent initialization captures the randomness in the training procedure, i.e., conditional on the observed data. I was a bit concerned how such training uncertainty relates to the uncertainty of data with respect to the true value. Intuitively, the uncertainty often refers to the data being unable to capture the truth because of lack of samples. One possible relation is that latter uncertainty is large in regions with scarce data, where the former training uncertainty will also be large. I Since this is related to the core theme of the paper, could you explain more on such relations, or on the uncertainty this work targets at?
The writing of the paper might be improved. I think it would be better to have fewer emphasized bold-font sentences, so that readers can focus on really important points. There are also some missing citations and references as well as typos in the paper which hurts the reading. I would expect the authors to have a careful check and make the presentation more organized.

There are some further questions:
What is the \pi in equation (3) in Section 4.2?
In theorem 5.2, the difference in ensemble variances is only discussed very vaguely. I don’t quite understand the meaning - it would be better if a direct comparison with equations is presented.
I am not sure whether the toy example and Figure 1 captures the advantage of MSG. Figure 1 shows the mean value and the standard deviations of the predictions, where the randomness can be viewed as from the training process, i.e., the random initialization, instead of the randomness in data. The crucial point should be that the estimations can provide a valid uncertainty quantification of the prediction with respect to the *true value*, so the shared LCB and shared min approach seems to directly target at the lower uncertainty bound (as observed in the paper). I would expect directly using the result of shared LCB/min can somehow provide a valid uncertainty quantification, without referring to the standard deviation induced by the training process. The approximation of the lower confidence bound should be more essential than evaluating the randomness in training process. Do you have more discussions on this issue?
Choice of hyper-parameters: In the experiment part, it would be better to add discussions on how the parameter beta and alpha are chosen. They seem to change across different domains and I think readers would expect some justifications on such choices (I only see ‘hyperparameter searching’, but not sure whether it’s cherry picking). Are the performances of both methods stable across different choice of hyper-parameters?

Minor points:
A missing citation in the introduction (page 1).
Missing “.” in Section 4.2.
Is there a missing superscript i for y(r,s’,pi) in equation (4)?
Missing reference in the caption of Figure 1.

**Summary Of The Paper:**

This paper proposes a novel method to utilize ensemble in quantification of uncertainty in offline RL, where the particular structure of dynamic programming and cumulative error are unique and interesting. Some theoretical justification emphasizes the ability of independent TD targets to capture the variance of the prediction, hence provide uncertainty quantification. Simulations show the proposed method gets larger variance in low-confidence regions. In experiments in benchmark environments, the proposed method outperforms those in the literature.

**Summary Of The Review:**

The proposed method to use independent initializations to capture the uncertainty is interesting, validated in simulations and performs well in experiments. However, the credibility of the work is compromised by the confusion in the type of uncertainty it targets at, which is the key theme. The writing could also be improved by another check to fix typos and missing references.

---

> ### Author Response · Authors · 2021-11-23
> **Author Response**
>
> Thank you for your detailed review of our work!
>
> **Form of Uncertainty**
>
> * This is a common question with another reviewer as well. We have thus added a new section, Appendix H, hoping to characterize the uncertainty we are estimating.
> * Following the discussion in appendix H, you are correct that we are in a sense capturing randomness in the training procedure due to random initialization. Our argument is as follows: Given an offline dataset and policy to evaluate, when we decide on a policy evaluation algorithm (e.g. Fitted Q Evaluation) and choose the network architecture, we are implicitly imposing a belief distribution over the Q-functions. In MSG we asked, given the imposed distribution over Q-functions, how can we optimize the policy in an offline RL setup?
> * Relation to uncertainty due to lack of data: Your intuition is correct. When there is a lack of data around a particular $(s,a)$, it is likely that different Q-functions in the ensemble make different predictions. As indicated by the Theorem in the manuscript as well as experiments in the Toy setting, we observe that such differences are propagated due to dynamic programming. As a result, along trajectories that -- under the policy being evaluated -- lead to sparse data regions, the uncertainty increases.
> * Relation to true values: There is generally no guarantee that the imposed belief distribution assigns high probability to the true Q-function (or even that the true Q-function is in the support of the belief distribution). However, if the neural networks are sufficiently expressive, as the size of the offline dataset increases, then distribution would be concentrated around the true Q-function.
> * We would like to emphasize that our notion of uncertainty is essentially identical to the notion of uncertainty for ensembles in supervised learning. Since ensembles are currently the most successful method for both accuracy and calibration [1], it motivated us to focus on effectively leveraging ensembles for offline RL.
>
> **$\pi$ in CQL-style regularizer equation**
>
> * We have updated this equation to be more clear. $\pi$ refers to the current policy.
>
> **Theorems and Toy example**
>
> * We have significantly revised the entire section “Independence in Ensembles Matters”.
> * We have replaced the two theorems with one theorem that directly compares the lower-confidence bound we obtain if we use “Independent” vs. “Shared-LCB”. We have also updated the accompanying text to the theorems.
> * Empirically, we emphasize that the implementation for "Independent", "Shared-LCB", and "Shared-Min" differ in only 2 lines of code, and the same hyper-parameter search budget was used. Yet, as shown in our results, Shared-LCB and Shared-Min already performed worse on D4RL gym subset, and completely failed on the harder antmaze tasks.
>
> **Hyperparameters**
>
> * As discussed in the appendix, we typically first do a coarse hyperparameter search using 2 random seeds. Based on the evidence we gather, we then choose a single hyperparameter and run it with 5 new random seeds to generate results for the Tables and Figures. Often our initial hyperparameter search is less extensive than described in appendix C.1. For example in appendix E.1 we describe that for RL Unplugged the initial hyperparameter search step used 8 possible combinations of $\beta, \alpha$. There are also general rules of thumb that we use when faced with a new domain. For a given new domain, we would first use low $\beta$ values: $\beta \in {-4, -8}$. If $\beta = -4$ is clearly better than $\beta = -8$, then we would guess that in this new domain high pessimism may not be necessary and explore the use of $\beta \in \{0, -1, -2\}$. For the $\alpha$ hyperparameter, we found that $\{0, 0.1, 0.5, 1.0\}$ is a wide enough range to explore. If $\alpha = 1.0$ was clearly better than the other values, then this would indicate to us that maybe the offline dataset is narrow (lacks diversity, e.g. imitation learning datasets) and then we could increase the value of $\alpha$. We have incorporated this practical advice for hyperparameter tuning, as well as plots demonstrating robustness of MSG in appendix section I.
>
> Thanks you!
>
> [1] Ovadia, Yaniv, et al. "Can you trust your model's uncertainty? Evaluating predictive uncertainty under dataset shift." arXiv preprint arXiv:1906.02530 (2019).

---

### Official Review · Reviewer_FzHF · 2021-11-02

**Correctness:** 2
**Technical Novelty And Significance:** 2
**Empirical Novelty And Significance:** 2
**Recommendation:** 3
**Confidence:** 2

**Main Review:**

The paper looks interesting, so are the approach, the theory, and observations.

# Revisions/questions/remarks
  * Roughly half of the papers refer to non peered reviewed papers (arXiv)
  * Hasn't the D4RL dataset been updated since Kumar et al. publication (CQL) which could explain the differences observed by the authors?
  * Lack of rigor in the theorems: the authors should have a formal theorem followed by its interpretation on which they base their intuitions
  * Lack of rigor in the results presentation: what bold format means? Maybe add a statistical significance test.
  * Too many claims not well supported
  * Section 5.2: wouldn't it have more impact if you used the git attached to Lee et al. 2019 in Section 5.2 to match the framework the authors are working on?

# Some typos
Lot of typos have to be corrected, here are some
  * p.1: "confident", citation missing
  * p.2: "data(Guo et al., 2017)" (missing space), "only has", "success"
  * p.5: Eq. 4 missing "i", "in the offline dataset. from the dataset."
  * p.8: footnote of p.7, "otherwis The"
  * p.9: "computationl efficieny"

**Summary Of The Paper:**

The paper propose to study ensemble methods for offline RL in the NTK framework. They derive

**Summary Of The Review:**

I found the paper quite hard to follow and overall there is a lack of rigor. However, looking at the experiments, there might be something to explore.

---

> ### Author Response · Authors · 2021-11-23
> **Author Response**
>
> Thank you for your review.
>
> * Roughly half of the papers refer to non peered reviewed papers (arXiv): Most of our citations refer to published peer-reviewed work. The reason the Arxiv versions are cited is because we took Bibtex citations from Google Scholar which happened to be Arxiv citations.
> * D4RL version: All of our results use the v0 version of the D4RL benchmark which is identical to the one used by CQL and the numbers reported in the D4RL whitepaper.
> * Theorems: We have replaced Theorem 4.1 with a new theorem, with a new following text.
> * Results Presentation: Bold format means that $mean - std$ is better than the $mean + std$ of any other reported baseline. Hence it is significantly better than the non-bold ones.
> * Too many claims not well supported: We are unsure of which claims you are referring to. We have noticeably improved many sections of the writing which we hope can address some of your concerns.
> * Section 5.2, NTK framework: We have added a new appendix section G.2 with a number of additional experiments. Notably, these results use the Neural Tangents library [1] to compare finite network results with infinite NTK network results based on our closed-form computations in appendix F.
>
> [1] Novak, Roman, et al. "Neural tangents: Fast and easy infinite neural networks in python." arXiv preprint arXiv:1912.02803 (2019).
>
> Thank you!

---

### Official Review · Reviewer_QFka · 2021-11-02

**Correctness:** 3
**Technical Novelty And Significance:** 2
**Empirical Novelty And Significance:** 3
**Recommendation:** 5
**Confidence:** 2

**Main Review:**

Strengths. The algorithm in this paper is model-free, which means it can be applied to realistic settings.
It used

Weakness. The theory part of this paper seems unfinished. The detailed derivation of the std should be included in the proof, and there should be more details, like descriptions of the network initialization,  in the description of the theory part

Typo in Line 8 of page 4. "This an ..."

**Summary Of The Paper:**

This paper provided a model-free algorithm for offline-RL. It provided Theorem 5.1 to justify its uncertainty quantifiers. It also conducted experiments to validate its algorithm.

**Summary Of The Review:**

Although this paper proposed an algorithm that can be applied to real-life applications without imposing strong structural assumption to allow uncertainty quantification, the theory part of this paper should be improved to be more complete, rigorous and self-contained.

---

> ### Author Response · Authors · 2021-11-23
> **Author Response**
>
> Thank you for your review!
>
> * We have replaced our theorem with a new one, as well as new accompanying text. Notably, the new theorem (and appendix F) directly computes and compares the lower-confidence bounds between Independent and Shared-LCB.
> * The network initialization is Glorot Normal. We will include this in the appendix.
>
> Thank you!

---

### Official Review · Reviewer_dnQ3 · 2021-11-04

**Correctness:** 2
**Technical Novelty And Significance:** 2
**Empirical Novelty And Significance:** 3
**Recommendation:** 5
**Confidence:** 5

**Main Review:**

The title of the paper: why so pessimistic? Is a bit little misleading. Actually most RL exploration methods are being the opposite: optimistic.

One major motivation of the reviewed paper is to “some of the unique challenges of uncertainty estimation in reinforcement learning ”. Could be explicit on this? The message/answer is not super clear after reading the paper. The DLTV paper has an observation that there are both intrinsic and parametric uncertainties going on with DRL. This shows the exploration for DRL is very challenging.  The DLTV paper uses distributional RL to estimate the upper confidence and lower confidence bound from the distribution of Q values, also using neural networks.

The presentation of the paper can be improved. Some times buzz words just come out without defining, like “deep ensembles”, "efficient approximation to ensembles"....

The results are evaluated on two ant mazes. The CQL paper performed pretty bad from Table 2. Did you try the other baseline? With only baseline and it’s poor, it is hard to see the proposed algorithm is strong.

Questions:
What is the connection of using ensembles and distributional RL for uncertainty estimation? In a superficial sense, both use a number of agents. The only difference is how they form an uncertainty estimator. However, identifying their connections and differences are interesting, isn’t it? For distributional RL, e.g., see the DLTV algorithm:
https://arxiv.org/abs/1905.06125

Actually the method proposed in this paper (estimating LCB, see Section 4.1) is very similar to DLTV. The formula is almost the same. Compare the equation above Sec. 4.2 with eq 4 of the DLTV paper. The only difference is that this is implemented using ensembles instead of distributional agents.

The statement “update of each member’s update is completely independent from each other”. Here some clarification is necessary. By independence, do you mean the learning rule sense? Do you have shared parameters between the members?

Beyond Method 1 and Method 2, have you thought about using shared maximum of the ensemble? See eq 8 of the ACE paper for example:
https://arxiv.org/abs/1811.02696
When coming to ensembles, the crucial question is what operator to choose to combine these estimate. The use of mean or max is a fundamental question to investigate.



Minor:
Some reference links are missing.

Refs:
LSPI
https://www.jmlr.org/papers/volume4/lagoudakis03a/lagoudakis03a.pdf

LAMapi:
https://sites.ualberta.ca/~szepesva/papers/lamapi.pdf

RKHS:
https://arxiv.org/abs/1206.4655

Pseudo-MDPs:
https://sites.ualberta.ca/~szepesva/papers/ieee_adprl2014.pdf

**Summary Of The Paper:**

There is a good literature for off-line RL. However, this  is reviewed at all. For example, LSPI (model free), LAMAPI (model-based), pseudo-MDPs (model-based) and RKHS embedding (model-based). The paper went straight to discussing the reduction of the value estimation error constraint. However, most of the algorithms use the error to directly learn the parameters. It is not necessarily formalized as constraint.

**Summary Of The Review:**

The paper addresses exploration with DRL algorithms, and explores estimating confidence interval of Q values using ensembles. Some discussion on the use of ensembles, the connection and difference from distributional RL, and comparisons to a good baseline (set) can make the paper much better and well supported.

---

> ### Author Response · Authors · 2021-11-23
> **Author Response**
>
> Thank you for your response!
>
> We believe you may have misinterpreted the topic area of our work. Our work is focused on the problem of "Offline RL", where a dataset trajectories in the MDP are provided, and we are no longer allowed to interact with the MDP. In contrast, your comments are regarding the problem of "Exploration in RL".
>
> Please find our responses to your comments below:
>
> * "Deep Ensembles" and "Efficient Ensembles" refer to terminology that have been established in the supervised learning literature [1]. "Deep Ensembles" refers to each ensemble member being a separate neural network. "Efficient Ensembles" refers to various approaches for making "Deep Ensembles" more efficient, for example by sharing parts of the networks.
> * Prior to our work, CQL has reported the best results of any prior work that we are aware of. Other prior methods have not been able to make any progress on the antmaze tasks. For this reason we chose it as the baseline to compare to CQL.
> * In our updated manuscript Appendix H we have included a description of the type of uncertainty we are estimating. With regards to your question, in our work we are focusing on epistemic uncertainty, whereas distributional RL would focus on aleatoric uncertainty (e.g. uncertainty due to stochasticity in MDP dynamics). We would imagine that our proposed method MSG could be combined with distributional RL methods to also incorporate aleatoric uncertainty. However, we do note that the benchmarks we used are in physical simulation domains, hence the MDP has deterministic transitions and there is little aleatoric uncertainty.
> * In "Deep Ensembles" there are no shared parameters between the ensemble members. Only the "Efficient Ensemble" approaches contain shared parameters. "Independent", "Shared-LCB", "Shared-Min", "Shared-Mean" refer to different manners in which we could compute the targets for each of the Q-functions (described in section 4.1 and 4.2). Independent refers to each Q-function using its own targets (which is the approach we motivated and take in MSG), whereas the "Shared" variants refer to all Q-functions using the same shared target values (for example, "Shared-Min" means that we compute the target values across the ensemble, take the minimum, and for all ensemble members use this minimum as the target values in the next policy evaluation procedure.)
> * We do not believe "Shared-Maximum" would lead to a good Offline RL method. In Offline RL it is necessary to be pessimistic, whereas "Shared-Maximum" would be optimistic.
>
> Thank you!
>
> [1] Ovadia, Yaniv, et al. "Can you trust your model's uncertainty? Evaluating predictive uncertainty under dataset shift." arXiv preprint arXiv:1906.02530 (2019).

---

### Official Review · Reviewer_XyHA · 2021-11-13

**Correctness:** 2
**Technical Novelty And Significance:** 1
**Empirical Novelty And Significance:** 2
**Recommendation:** 3
**Confidence:** 4

**Main Review:**

Offline RL is certainly an important area of research and there is still plenty to do, since the existing algorithms have all shown their limits. However I believe that the submission falls short on several crucial points that I enumerate below:
* There is a terminology confusion with the word pessimistic.
   * Offline RL inductive bias: Offline RL algorithms rely on the assumption that the control policy(ies) is(are) much better than average, and therefore that omitted actions are likely to be underperforming. This is a desirable feature, and the ex machina argument made by the authors that "it would be preferable if we could place more trust into the predictions of value networks beyond the training dataset" is not supported. And this is not what the authors propose to do: they simply propose a way to estimate uncertainty and to enforce this assumption: LCB means that the training is going to be conservative.
   * Pessimistic meaning in the offline RL literature: Usually, pessimistic refers to the opposite of optimistic exploration: methods that distort the reward/value function to induce pessimism in the face of uncertainty. It is generally used in opposition to methods that conservatively constraint the policy search [Petrik2016], [Fujimoto2019], [Laroche2019]. Most (all?) RL algorithm can be viewed as policy iteration: loop over policy improvement and policy evaluation. Pessimism refers to pressure on the policy evaluation step, and constraint of the policy search refers to pressure on the policy improvement step. The method developed in the paper applies pressure on the value function and is therefore pessimistic, hence the confusion for the readers. There is even a third approach consisting on early stopping the loop [Brandfronbrener2021].
* The novelty is limited and the positioning incomplete. As pointed out by the authors, the use of deep ensembles for offline RL is not novel. The authors claim that the networks share target. Usually yes, but not always: Algorithm 2 of [Osband2018] is training each network independently. And when they train them all together, they share a form of pessimistic target (reusing the paper terminology: independent double Q, shared LCB, shared min), not the expectation (shared mean), otherwise it does not make sense!
* The theoretical analysis is not convincing. Following my previous point, it would have made the analysis much stronger if the other forms of pessimistic updates were accounted for and analyzed. It is straightforward that using a common target will provide a collapse of the inter-network variance. The use of infinite width NN is a big hammer for such a small nail. Figure 1 reports an experiment but its result is inconclusive since all pessimistic updates deliver the same lower bound.
* The authors claim that shared LCB, shared min, CQL were unable to train effective policies, although these have been reported in the past. This, combined with the lack of compelling arguments on the theory side, cast doubt in the empirical results. Also, "the other data setting could not adequately differentiate between competitive methods" is not an argument for not reporting results.

[Petrik2016] Petrik, M., Ghavamzadeh, M., & Chow, Y. (2016). Safe policy improvement by minimizing robust baseline regret. Advances in Neural Information Processing Systems, 29, 2298-2306.

[Osband2018] Osband, I., Aslanides, J., & Cassirer, A. (2018). Randomized prior functions for deep reinforcement learning. arXiv preprint arXiv:1806.03335.

[Fujimoto2019] Fujimoto, S., Meger, D., & Precup, D. (2019, May). Off-policy deep reinforcement learning without exploration. In International Conference on Machine Learning (pp. 2052-2062). PMLR.

[Laroche2019] Laroche, R., Trichelair, P., & Des Combes, R. T. (2019, May). Safe policy improvement with baseline bootstrapping. In International Conference on Machine Learning (pp. 3652-3661). PMLR.

[Brandfronbrener2021] Brandfonbrener, D., Whitney, W. F., Ranganath, R., & Bruna, J. (2021). Offline RL Without Off-Policy Evaluation. arXiv preprint arXiv:2106.08909.

**Summary Of The Paper:**

The paper agues that methods, which they call "support constraint", consisting in assuming that any action outside the training yields negative outcomes, lead the agent's training to be overly pessimistic. In contrast, they propose MSG an algorithm where a lower bound of the value function is estimate from ensembles of networks. They provide some theoretical arguments on infinite-width neural networks and validate their methods on many applications, ranging from illustrative toy examples to classic continuous state-action spaces offline RL benchmarks.

**Summary Of The Review:**

I recommend reject for the reasons detailed in the full review.

If the authors could clarify the following things, I would consider raising up my score:
* How does their algorithm differs from Algorithm 2 of [Osband2018]?
* Why would the LCB of independent networks would have a better uncertainty estimate than an ensemble network learnt on common LCB targets?
* Why were the authors unable to train effective policies of well established algorithms?

---

> ### Author Response · Authors · 2021-11-19
> **Author Response**
>
> Thank you for your detailed constructive feedback regarding our work.
>
> Throughout our response we reference equation, theorem, figure, table, and section numbers based on the most recently updated manuscript revision (Nov. 19th)
>
> In addition to providing responses to your detailed comments, we hope our reply and updated manuscript provide direct responses to the key bullet points noted in your “Summary of The Review” section.
>
> **Terminology Confusion**
>
> * Offline RL Inductive Bias:
> ** We respectfully disagree with your claim that “Offline RL algorithms rely on the assumption that the control policy(ies) is(are) much better than average, and therefore that omitted actions are likely to be underperforming.” We believe a key use case of offline RL is that one can have very large datasets of agents performing a variety of task behaviors, and for a target task one could simply define a reward function (or condition on new goals, etc.) and train a new agent in an offline manner. In such scenarios, it could be that explicit attempts at the target task do not even exist in our dataset. A toy representation of this scenario is captured by the D4RL antmaze tasks: Since the data consists of the ant running in arbitrary directions, a majority of the dataset is not related to the trajectory the ant must take at test time (in antmaze tasks, the agent is being optimized to run from a specific corner of the maze to another specific corner). On these tasks, which arguably are much more representative of the challenge of offline RL (e.g. being able to stitch segments of different trajectories), our work MSG achieved unprecedented results. Our offline RL results on these domains even far exceed recent unpublished work on arxiv when they use additional online fine-tuning [4].
> ** We completely agree that LCB is going to be pessimistic. Please view our next response for why we believe LCB is a less pessimistic approach that prior work.
> * Pessimistic meaning in the offline RL literature:
> ** We would like to clarify that we completely agree that MSG is a pessimistic approach. However – using your proposed terminology – we also believe that MSG is a **less pessimistic** approach than what is used in the current offline RL literature, hence the title “Why so pessimistic?”.
> ** Our reasoning for why our approach is less pessimistic is as follows: Under the terminology of Buckman et al. 2020 [1], essentially all current SoTA offline RL methods (a variety of which are described in the second paragraph of the Introduction section) fall under “Proximal Pessimistic Algorithms”. This is because conservative policy constraint approaches, or pessimism approaches which push down on the values of out-of-dataset actions (such as CQL) correspond to an assumption that any action outside the dataset can lead to worst-case outcomes, which is a trivial uncertainty function (Buckman et al. [1] (section 5.3)). Instead, in our work we have aimed at transferring state-of-the-art uncertainty estimation techniques from supervised deep learning, and optimizing the policy with respect to the lower-confidence bound of derived uncertainties, thus relaxing the worst-case assumption made by the majority of work in offline RL with deep neural networks. The effect is that by estimating uncertainties we are avoiding explicitly encouraging the policy to remain close to the offline dataset; if the ensemble of Q-functions agrees that a particular action outside the offline dataset is good, then there is nothing preventing the policy from taking that action.

---

> > ### Author Response · Authors · 2021-11-19
> > **Author Response**
> >
> > **The novelty is limited and the positioning incomplete**
> >
> > * While similar to Osband et al. 2018, there are important differences:
> > ** Setting: Osband et al. 2018 is focused on exploration, while we are focusing on offline RL.
> > ** Algorithmically: It is not clear how one would convert Osband et al. 2018 into a performant offline RL method. In the exploration setting which they work with, they train independent Q-functions, and in each episode they choose one of the Q-functions to use for selecting actions. If we convert this method to offline RL, for each ensemble member we would have the same update $(Q^i(s,a) - (r + \gamma \cdot \bar{Q}^i(s’, a’)))^2$, where $a’ = \text{argmax}\bar{Q}^i(s’, \cdot )$. **There is no pessimism involved, and thus it would likely not be a good offline RL method.** In contrast, when we convert MSG algorithm from actor-critic to the Q-Learning setting of Osband et al. 2018, we get $a’ = \text{argmax LCB}(\bar{Q}(s’, \cdot ))$ (i.e. the argmax action of LCB is used by all ensemble members).
> >
> > * Novelty and Contributions:
> > ** Estimating uncertainties with deep networks is a very challenging and open problem. It is likely for this reason that uncertainty estimation techniques have thus far not been prevalent in the deep offline RL literature (a list of some we include below). A major contribution of our work is that we investigate the transfer of modern uncertainty estimation techniques in deep learning to offline RL: 1) we theoretically and through toy examples study key algorithmic choices, 2) we present very strong empirical results on established benchmark problems, 3) we demonstrate the shortcoming of “efficient” uncertainty estimation techniques when they are transferred from regression and classification to the domain of RL, thus highlighting very clear avenues for future research.
> > * * * In this extensive list of offline RL methods (https://github.com/hanjuku-kaso/awesome-offline-rl), we only found a handful of methods using deep networks and uncertainty estimation. EDAC [2] (discussed below), MOPO [8] and MBOP [9] (model-based methods using aleatoric uncertainty, not epistemic uncertainty), UWAC [10] (using dropout for computing variance (uncertainty) and downweighting Q-function update on (s,a) with high variance), POPO [11] (a BCQ [12]-style approach using obtaining pessimism from distributional RL, which corresponds to aleatoric uncertainty and not epistemic uncertainty).
> > ** For the state-of-the-art uncertainty estimation method “Deep Ensembles”, we have demonstrated through mathematical argumentation, toy experiments, and solid benchmark results why independence in ensembles matters, as opposed to the ensembling approach currently used by most of the literature.
> > ** In addition to deep ensembles which are considered to be the SoTA uncertainty estimation technique in the deep learning literature, we explored modern “efficient ensembles” and demonstrated a clear shortcoming, thus opening important avenues for continued research.
> > ** The empirical results of MSG with deep ensembles is undeniably superior to prior work, as far as they can be captured by current benchmarks. No prior work has been able to make nearly as much progress on the antmaze domains as MSG. To further support the strong empirical results of MSG, in our updated manuscript we have added new experiments using the RL Unplugged benchmark as well [5]. Using network architectures that have $\frac{1}{60}$ of the parameters as typically used in RL Unplugged, MSG rivals the best reported results on these domains (with the exception of humanoid run task, which through behavioral cloning experiments we observed really does require the exceedingly large network sizes).

---

> > > ### Author Response · Authors · 2021-11-19
> > > **Author Response**
> > >
> > > **The theoretical analysis is not convincing**
> > >
> > > * We would like to clarify a potential miscommunication here: We never intended to suggest that Shared-Mean would be a valid ensemble approach, and indeed in our benchmark comparisons we only tried Shared-LCB and Shared-Min as alternatives. We agree with your comments that the important theoretical comparison is between Independent and Shared-LCB. **Based on your valid criticism, we have now replaced Theorem 4.1 (and updated Appendix F), and now directly compare Independent with Shared-LCB, showing a key difference theoretically as well.**
> > > * While infinite-width networks are a big hammer to use, they also provide us with a big assumption to carry out our analysis, namely the behavior of the networks throughout training being linear.
> > > * We have also included a new Appendix section G.2 which in our opinion presents very intriguing empirical observations for future work to build upon. These observations demonstrate that while we can make interesting predictions using the NTK limit, other infinite-width parameterizations seem to lead to better uncertainties. Although the specific alternative parameterization we explored in Appendix G.2 does not have simple closed-form equations describing the evolution of the networks during training, this strongly motivates future investigations into these differences.
> > >
> > > **Baseline Results**
> > >
> > > * Shared LCB, Shared Min:
> > > ** While Shared LCB and Shared Min have been popular choices, especially in offline RL, we are not aware of any work where this alone was sufficient to obtain good offline RL policies. The closest works to “pure” Shared LCB and Shared Min that we are aware of are EDAC [2] and EMaQ [3]. In EDAC, Shared Min with a very large ensemble of Q-functions is proposed. However, they observed a lack of diversity in the Q-functions, and proposed “Ensemble Gradient Diversification” to combat this issue. In EMaQ, a strong autoregressive generative model is fit to the data, and Shared-Min backups are performed on actions sampled from this generative model. Thus, EMaQ has a very strong conservatism policy constraint.
> > > ** We emphasize that the implementation difference between Independent, Shared-LCB, and Shared-Min is only 2 lines. Also, we used the same hyperparameter search budget for all baselines, which further provides us with confidence regarding our results. Shared-LCB performs ok on gym locomotion tasks, but when transferred to much harder antmaze tasks, it completely fails. Shared-Min generally always performed poorly; the likely reason is that as the ensemble size increases, the minimum values becomes less and less thus hurting the uncertainty estimation.
> > > * CQL: A number of papers have reported issues with reproducing CQL in D4RL domains. Examples include: F-BRC [6] Table 1, TD3+BC [7] Footnote 4 and Table 4, EDAC [2] Page 9 top. Also, please note that our implementation of CQL performed very well on the gym domains of D4RL (often significantly better than prior reported values), but once transferred to the antmaze domain it almost completely failed. Thus to be more fair, we reported the number presented by the original CQL paper.
> > > * "the other data setting could not adequately differentiate between competitive methods": From the gym subset of D4RL, the setting we omitted were “random”, “medium-expert”, and “expert”. The “random” dataset is generated by a random policy, hence the ceiling of performance is very low. The “expert” dataset is effectively an imitation learning dataset, and the “medium-expert” dataset is the concatenation of “medium” and “expert” datasets. Colleagues of ours have seen that by training a small neural network binary classifier, they can differentiate between “medium” and “expert” with 99% accuracy, meaning the datasets effectively differ only on the first few states of the trajectory. These aspects limit the value of these omitted data settings for evaluating offline RL. On the “medium” and “medium-replay” settings (which contain much broader diversity of data), as discussed in the paper we heavily tuned MSG and the baselines (with equal budget) and as reported we still could not see a meaningful difference between MSG, CQL, and F-BRC. Thus, we did not attempt the computationally expensive tuning on “random”, “medium-expert”, and “expert”. If given our reasoning you believe these experiments would still be necessary, please let us know and we will try to generate the results prior to the author discussion deadline.

---

### Comment · Area_Chair_YhPk · 2021-11-10
**questions for authors**

After looking at the paper and the reviews, I have a few questions for the authors -- it would be great if the author response could address any of these, thanks!

Can you describe the statistical model that we are attempting to approximate? I.e., if there were no computational limits, how would we specify a prior over the MDP and/or its value function, how would we compute the posterior (analogous to Fig 1 left), and what would the posterior uncertainty look like?

In the limit of increasing computation (i.e., increasing ensemble size and increasing number of random features in each ensemble member), is the proposed approach exact? (I.e., does it match the ideal unlimited-computation model from the previous paragraph?) Can we characterize any departures from exactness?

The chain MDP example in sec 5.2 is small enough that we shouldn't have to worry about computational limits. Can we compute the exact posterior, at least numerically, and compare to the experimental results?

An interesting special case is contextual bandits (i.e., horizon 1). In this case there is no issue of propagation of uncertainty, so we're much more similar to the supervised setting. In this case, at least, it appears that the proposed approach is not exact: e.g., if we train to completion, every ensemble member will interpolate the training points (the observed (s, a, r) tuples), while the true posterior will still have residual uncertainty even at the training points. Is the above worry accurate, and is there anything we can do to get a better representation of the posterior?

Given that ensembles are being used already, it would be computationally nearly free to add bootstrapping (passing only some of the training data to each ensemble member); the cited paper of Osband et al. is an example of this. Would some sort of bootstrapping help the issue raised in the previous paragraph?

One common problem with value function approximation in RL is bias due to the nonlinearity of the max operator: we tend to prefer actions where the function approximator happens to be over-optimistic. This problem affects different RL methods differently; e.g., DQN suffers badly, while actor-critic is less affected. The effect for the current algorithm seems like it would be that we'd get a different "effective beta" compared to the one we set -- with beta unfortunately varying over the input space. Is the above description accurate, and is there anything we can do to improve the situation?

While the NTK approach is one of the best ways we have to understand the behavior of deep nets, it is not a perfect model of deep nets as used in practice. Can we say anything about how departures between NTK and in-practice deep nets would affect the proposed method?

---

> ### Author Response · Authors · 2021-11-19
> **Author Response**
>
> Thank you for your thoughtful feedback and detailed questions regarding our work!
>
> Throughout our response we reference equation, theorem, figure, table, and section numbers based on the most recently updated manuscript revision (Nov. 19th)
>
> **The statistical model we are trying to approximate**
>
> Short Answer:
> The statistical model that we are attempting to approximate is not a Bayesian one; the uncertainty estimation technique we employ is Deep Ensembles, and generally it is not considered to be a Bayesian form of uncertainty estimation. In our work, we have not explicitly computed the distribution we are trying to approximate, but we have characterized various properties of distribution on which our proposed approach depends on. Based on the equations in Appendix F, the distribution over converged Q-functions is a Gaussian Process (GP) whose mean and covariance functions depend on 1) the architecture, 2) the weight initialization distribution, and 3) the data. In appendix section F.5, we compute in closed form the mean and marginal variances of this GP for our proposed approach Method 1/Independent Targets/MSG. However, we did not attempt to compute the covariance function of the GP.
>
> Detailed Response:
>
> This is an interesting question, and it would be useful to break the answer into two parts.
>
> First, let us ignore that we are in the RL setting. Consider the following mean squared error regression problem: We are provided with a dataset $\{(x, y)\}$, and we want to train an infinitely wide neural network $f$ to fit this dataset using mean squared error loss. In Lee et al., 2019 [1] it is shown that the distribution – randomness due to random initialization – of converged functions is given by the Gaussian Process (GP) in equation 16 of their work. They note however that this GP does not have the interpretation of a Bayesian posterior. In this response let us call this GP, GP-nobayes. We are aware of one work by He et al., 2020 [2] which proposes modifications to the function $f$ such that the GP corresponding to the distribution of converged solutions does have a Bayesian interpretation, which here we denote by GP-bayes. In our work we did not investigate using the modifications of [2] for obtaining the GP-bayes equivalent for value function learning. Nonetheless, GP-nobayes (equation 16 of [1]) has an interesting mean and covariance structure: they are functions of the Neural Tangent Kernel (NTK) [3] and another kernel (equation 12 of [1]), both of which depend only on the architecture of the neural network and the random initialization. Hence, by designing specialized architectures and weight initialization schemes, we are effectively designing the GP-nobayes distribution.
>
> When we move to the RL setting where we are estimating the value functions, we find the situation to be a bit more complex. In Method 1 (Independent Targets, As in our work MSG), after each iteration, we compute the target value $y^i$, and each ensemble member uses its own targets in the next iteration. Based on the equations in appendix F, the distribution over converged Q-functions is a Gaussian Process (GP) whose mean and covariance functions depend on 1) the architecture, 2) the weight initialization distribution, 3) the data. In appendix F.5, we compute in closed form the mean and marginal variances (i.e. Var[Q(s,a)]) of this GP for our proposed approach Method 1/Independent Targets/MSG. However, we did not attempt to compute the covariance function of the GP.
>
> In the cases where the targets are shared, after each iteration of fitting the Q-functions we compute the target values $y^i$ for each ensemble member, and use a shared $\bar{y}$ as the targets for all of the Q-functions in the next iteration. Because all ensemble members use the same target, in each iteration we are effectively solving a new mean-squared error regression problem. So the resulting distribution will have the same form as GP-nobayes. We believe it is for this reason that in Figure 1, Shared Mean, Shared LCB, and Shared Min only have randomness in the region with no data, because they behave as if they are solving a regression problem where an oracle provided the final shared target values $\bar{y}$ to use.
>
> **Based on comments from Reviewer XyHA, we have replaced Theorem 4.1 with a new Theorem which directly compares the lower-confidence bound (LCB) of MSG to Shared-LCB setting.** We hope this also provides a more accurate picture of what our approach is estimating.

---

> > ### Author Response · Authors · 2021-11-19
> > **Author Response**
> >
> > **Is the proposed approach exact**
> >
> > Yes. We believe the proposed approach is exact.
> >
> > In the previous section we mentioned that the distribution of Q-functions after $t$ iterations of Fitted Q Evaluation is a Gaussian Process (GP) whose mean and marginal variance are characterized by appendix section F.5. Hence, each ensemble member after $t$ iterations is a sample from this GP. As a result, with finite ensemble size we are performing a Monte-Carlo approximation of this distribution. With regards to how close the results of finite networks are to infinite networks, please refer to our response to your next comment below.
> >
> > **Exact computation in toy setting**
> >
> > We have included new results in appendix section G.2. Summary: As discussed there, the Toy experiments in the main manuscript did not use the NTK network parameterizations and gradients descent, but instead used the standard neural network parameterization and Adam optimizer as we thought they would be more reflective of practical settings. In appendix G.2 we present two types of results. First, we demonstrate that when the NTK parameterization and gradient descent are used, as the network width increases, the results appear more and more similar to our closed-form infinite-width calculations. Second, these experiments demonstrate that with the NTK parameterization, as the width increases, the uncertainty structure appears less useful. We connect this to recent work presenting an infinite-width parameterization that learns data-dependent features, and shows that empirically they resolve the observed problem. Unfortunately, this parameterization does not have closed forms equations that we can readily use as we did in the NTK settings, but they hint at very interesting avenues for future theoretical work.
> >
> > **Contextual Bandits**
> >
> > In the case of contextual bandits, there will be residual uncertainty at the training points only if the reward is stochastic. While this is an interesting consideration, in this work we always assumed deterministic rewards and will make this point more clear in the manuscript. In the case of deterministic rewards we believe our proposed approach would be exact in the contextual bandit setting as well.
> >
> > **Bootstrapping**
> >
> > As described in the response to the previous question, we believe the connection to the horizon 1 case of contextual bandits is about deterministic vs. stochastic rewards, so we do not think bootstrapping would address the concern.
> >
> > In terms of increasing the diversity amongst the ensemble members, in [5] where ensembles were used for exploration it was observed that bootstrapping did not result in a noticeable advantage and that simply the random initialization of the networks provided sufficient diversity.  We have also tried randomly weighting the losses in the mini-batch, but we did not see a noticeable difference whether we used this or not. i.e. if $L$ represents the vector containing the Q-function losses for each mini-batch element, we multiplied $L$ element-wise with a random vector $U$ whose elements contained samples $u \sim Uniform[0, 2]$.
> >
> > **Bias due to maximization in RL**
> >
> > With respect to your comment on “effective beta”, $\beta$ is a constant hyperparameter. Did you mean that because there is a preference for over-optimistic actions, then the variances for these actions could somehow be affected? Would you be able to further clarify this question?
> >
> > In our experience working with MSG, we have had an empirical observation that may be relevant to this question. When we use very low ensemble sizes (e.g. 1, 2, 4), or in some offline RL benchmark domains when we set $\beta = 0$, we see that the Q-functions “explode”; we see that the magnitude of predicted values tends to infinity and so do the Q-function losses. However, when we use a large ensemble size (we used $N=64$ for almost all experiments in this work), and use $\beta < 0$, we never observe such instabilities. As the Q-function losses are standard L2 backups, we believe this indicates that the MSG policy optimization objective (i.e. gradient ascent on the lower-confidence bound of Q-values) does a very good job at mitigating over-estimation biases often observed in RL (particularly offline RL) and prevents the policy form exploiting individual Q-functions.

---

> > > ### Author Response · Authors · 2021-11-19
> > > **Author Response**
> > >
> > > **Theory vs. Practice**
> > >
> > > While there exists a literature characterizing deviations from infinite-width regime when using finite networks [6], we are not intimately familiar with this literature and have not considered its effects. As discussed above, we have included new toy experiment results in appendix section G.2 towards obtaining a better understanding of wide networks. Our empirical results suggest that as the network width increases, the uncertainties obtained from NTK parameterization may become less useful. Thus, we investigated experiments with Maximal Parameterization [7] of wide networks which maintain useful uncertainty structure even at very wide widths.
> > >
> > > Overall, we agree that transferring our theoretical presentation to practice requires handling a number of important factors such as, Adam optimizer instead of gradient descent, using mini-batches instead of full-batch learning, stochastic policy instead of a deterministic policy (although this point was more due to notational convenience and simplicity of equations), alternating gradient steps between critic and policy updates instead of fully fitting critics in each iterations, finite width networks, and finite ensemble size. Nonetheless, our theoretical results are corroborated by the toy experiments as well as clear difference in benchmark results between Independent (MSG) and Shared-LCB ensembles.
> > >
> > > **Additional Benchmark Experiments**
> > >
> > > In our updated manuscript, we have added new experiments using the RL Unplugged benchmark as well [8]. Using network architectures that have 1/60 of the parameters as typically used in the domains, MSG rivals the best reported results on these domains (with the exception of humanoid run task which through behavioral cloning experiments we observed really does require the exceedingly large network sizes).
> > >
> > >
> > > Thank you for the very interesting questions!
> > >
> > >
> > > Citations:
> > >
> > > * [1] Lee et al., 2019
> > > * [2] He, Bobby, Balaji Lakshminarayanan, and Yee Whye Teh. "Bayesian deep ensembles via the neural tangent kernel." arXiv preprint arXiv:2007.05864 (2020).
> > > * [3] Jacot, Arthur, Franck Gabriel, and Clément Hongler. "Neural tangent kernel: Convergence and generalization in neural networks." arXiv preprint arXiv:1806.07572 (2018).
> > > * [4] Novak, Roman, et al. "Neural tangents: Fast and easy infinite neural networks in python." arXiv preprint arXiv:1912.02803 (2019).
> > > * [5] Osband, Ian, et al. "Deep exploration via bootstrapped DQN." Advances in neural information processing systems 29 (2016): 4026-4034.
> > > * [6] Hanin, Boris, and Mihai Nica. "Finite depth and width corrections to the neural tangent kernel." arXiv preprint arXiv:1909.05989 (2019).
> > > * [7] Yang, Greg, and Edward J. Hu. "Feature learning in infinite-width neural networks." arXiv preprint arXiv:2011.14522 (2020).
> > > * [8] Gulcehre, Caglar, et al. "RL Unplugged: A Suite of Benchmarks for Offline Reinforcement Learning." Advances in Neural Information Processing Systems 33 (2020): 7248-7259.

---

> > > > ### Comment · Area_Chair_YhPk · 2021-11-22
> > > > **follow-up**
> > > >
> > > > ### Statistical model
> > > >
> > > > Apologies for the confusion! I think this question may have been obscured by my use of Bayesian terminology (prior, posterior, etc.). I agree that there's no need to pick Bayesian statistics as our formalism; if we start from different axioms we can arrive at different reasoning systems, including for example frequentist statistics or conformal prediction. What I was intending to ask was, in whatever formal reasoning system we'd like to use, what is the ideal answer, given access to arbitrary computational resources, so that approximations are unnecessary? I.e., how do we quantify our uncertainty about the MDP and value function before seeing data, and how do we quantify it after seeing data?
> > > >
> > > > Importantly, neither Gaussian processes nor deep ensembles can be the answer to this question: they both make approximations that are only necessary for computational reasons. (GPs approximate by pretending that the distributions of interest are all Gaussian, which they clearly are not in the case of MDP Q functions; ensembles approximate by representing general distributions using finite samples.)
> > > >
> > > > Hopefully the intended question is clearer from the above? Sorry for the confusion from my earlier phrasing!
> > > >
> > > > ### Exactness
> > > >
> > > > Here I was intending to ask for a comparison to the exact distribution (the one described above). Since the example of contextual bandits from my first post depends on stochastic rewards, which are outside the paper's assumptions (thanks for clarifying this!), here's another example that might shed light on this question: consider a horizon-2 MDP, where the states at the two time steps are distinct (i.e., the learner always knows how many steps are remaining). As you point out, the GP will give a reasonable answer at the final time step: it will predict zero uncertainty at the training points, and Gaussian uncertainty at unobserved (s, a) pairs. But at the next-to-last time step, there seems to be a problem: the true backed up distribution will be a mixture of Gaussians, while the GP will necessarily tell us that there is Gaussian uncertainty in the Q-value at every state. (In the backed up distribution, the mixture components will correspond to the actions that the policy selects and the outcomes we obtain (which s' we reach for a given (s, a)); both of these sources of uncertainty can be non-Gaussian, e.g., discrete.)
> > > >
> > > > The results in Appendix G.2 are definitely interesting, thanks! But, they seem to answer a somewhat different question, namely the effect of approximation of the NTK by finite width networks.
> > > >
> > > > ### Bias due to maximization
> > > >
> > > > Sorry for the confusion in my original phrasing! Yes, exactly, in RL algorithms in general, the bias and variance at each state-action pair could be affected by the preference for over-optimistic actions; and different state-action pairs could be affected differently. The phrase "effective beta" was intended to mean that the calculated LCB could actually correspond to a different number of (true) standard deviations away from the (true) mean, due to this effect on the estimated mean and standard deviation.
> > > >
> > > > After looking at the additional information provided in the author response, I think I can understand better how this issue will affect the MSG algorithm: within the first policy evaluation step, there will be no bias from this issue, since we are not attempting to change our action selection. When we update our policy, there will be bias to pick over-optimistic actions. When we evaluate the updated policy, the effect is more subtle: since we're again using the same sample of state-action pairs, there's some dependence between the policy update and the new learned Q-function; but it's unclear exactly what the effect of this dependence will be. Is the above correct, and can you provide any additional intuition for what will happen?
> > > >
> > > > ### Bootstrapping
> > > >
> > > > It might be worth re-answering this question given any revisions based on the above?

---

> > > > > ### Author Response · Authors · 2021-11-23
> > > > > **Author Response**
> > > > >
> > > > > Thank you for your follow-up response!
> > > > >
> > > > > **Statistical Model**
> > > > >
> > > > > We hope our new appendix section H can provide a response to this question.
> > > > >
> > > > > **Exactness**
> > > > >
> > > > > * Ensembles being exact: As described in our new appendix section H, conditioning on data + algorithm (Fitted Q Evaluation) + architecture choice + policy imposes a belief distribution over the Q-functions. Thus ensembles are exact Monte-Carlo samples of this distribution.
> > > > > * Horizon 2 MDP: As described in our new appendix section H, the object we are trying to approximate and be pessimistic about is $Q^\pi(s, \pi(s)) = r(s,a) + \gamma \cdot E_{s’ \sim MDP, a’ \sim \pi(s’)}[Q^\pi(s’, a’)]$. Instead, we believe your question pertains to a setting where we are estimating a distribution over returns, e.g. $G(s_t,a_t) = r(s_t, a_t) + \gamma r(s_{t+1}, a_{t+1}) + \ldots$. The difference is that in our approach we are estimating the epistemic uncertainty about $Q^\pi_\theta(s, \pi(s))$ whereas in your example we would be estimating the aleatoric uncertainty due to unknowable random outcomes in the MDP transitions. In our work we have not touched upon aleatoric uncertainty (e.g. using distributional RL algorithms) and all our experiments were in physical simulation benchmarks where the dynamics are deterministic. We envision that MSG could readily be applied to distribution RL methods which would be very valuable for domains such as medicine which do have randomness in transition dynamics.
> > > > >
> > > > > **Bias due to maximization**
> > > > >
> > > > > * We would imagine that such effects do occur, and it is a problem that likely plagues most algorithms that interleave policy evaluation and policy optimization (i.e. most (all?) RL algorithms). There exists some work aimed at reweighting the distribution of state-action pairs used for updates [1], but we did not employ such mechanisms.
> > > > > * Intuition for the interplay of policy optimization and policy evaluation: In a standard actor-critic setup, in the policy optimization step the policy will try to pick actions for which the Q-function is over-optimistic. In MSG however, we have an ensemble of N Q-functions, **so the exploitation effect will only happen if all the Q-functions are over-optimistic about the same action**. Intuitively if some ensemble member is over-optimistic about an action, some of the other ensemble members might not be, thus increasing the standard deviation of the prediction, and resulting in increased pessimism in the lower-confidence bound (LCB). And then in the next iteration, because the policy does not exploit the Q-functions, their updates will be stable, resulting in a fortuitous loop. Our empirical experience working with MSG also suggests that exploitation does not happen with a sufficient ensemble size; as discussed in our previous response, we observed that with very small ensemble size (e.g. 1, 2, 4) the training of Q-functions can diverge with their predictions tending to infinity (typical symptoms of policy exploitation). But with large ensemble sizes, and with low beta values (e.g. -8), we rarely see such instabilities in training.
> > > > >
> > > > > **Bootstrapping**
> > > > >
> > > > > Practically speaking, in the deep learning literature many works have reported not seeing an advantage from using bootstrapping. In Bootstrap DQN [3], it is reported that there is no difference whether bootstrapping is used or every ensemble member uses the entirety of the data. In [2], detailed experiments in supervised learning setting demonstrate that ensembles perform better with respect to accuracy and calibration when every ensemble member uses the entirety of the dataset. Thus, we do not expect to gain much advantage from the use of bootstrapping in MSG.
> > > > >
> > > > > Thank you!
> > > > >
> > > > >
> > > > > [1] Kumar, Aviral, Abhishek Gupta, and Sergey Levine. "Discor: Corrective feedback in reinforcement learning via distribution correction." arXiv preprint arXiv:2003.07305 (2020).
> > > > > [2] Nixon, Jeremy, Balaji Lakshminarayanan, and Dustin Tran. "Why Are Bootstrapped Deep Ensembles Not Better?." ''I Can't Believe It's Not Better!''NeurIPS 2020 workshop. 2020.
> > > > > [3] Osband, Ian, et al. "Deep exploration via bootstrapped DQN." Advances in neural information processing systems 29 (2016): 4026-4034.

---

> > > > > > ### Comment · Area_Chair_YhPk · 2021-11-23
> > > > > > **exactness**
> > > > > >
> > > > > > "conditioning on data + algorithm (Fitted Q Evaluation) + architecture choice + policy imposes a belief distribution over the Q-functions": I'm not sure I understand this statement. Is the intent to say that there's some effective prior built into the algorithm, architecture, etc., so that when we run the algorithm on some data we get (an approximation of) the corresponding posterior? (Or a similar statement in some non-Bayesian formalism?) If so, it's not obvious that this is true; in general there's no reason for an algorithm to behave this way, even if the algorithm is phrased in terms of operations on distributions. For example, some implicit assumptions of an algorithm could be false, or some constraints imposed could be mutually inconsistent, so that no possible prior can explain the behavior of the algorithm. In the current paper, for example, the following assumptions seem mutually inconsistent: (1) that the vector of all Q(s,a) is jointly Gaussian, and (2) that the vector of all Q(s,a) follows the Bellman equation. (In particular, the set of Gaussian distributions is not closed under the max over actions that appears in the Bellman equation.)
> > > > > >
> > > > > > It's possible that this inconsistency, and any others that might exist, are benign; but it's also possible that inconsistencies such as this one could lead to problems -- e.g., converging to the wrong answer as we get more data, or even becoming unstable. This possibility is the motivation for asking the original question of what our ideal answer is.
> > > > > >
> > > > > > "In this work, we do not quantify uncertainties about any aspect of the MDP itself": it's unclear how we could be sure to avoid inconsistencies (and the resulting problems described above) without attempting to make the correspondence between the behavior of the algorithm and our uncertainty about the underlying MDP.
> > > > > >
> > > > > > To be clear, it seems likely that the way GPs are used in this algorithm is benign: the assumptions behind GP regression are not formally satisfied, but the behavior of GP regression is likely robust to this mismatch. But it would be preferable to explicitly call out this mismatch and discuss it, and argue that there won't be problems -- else a reader might mistakenly assume that the paper is claiming that the derivation is airtight rather than heuristic.
> > > > > >
> > > > > > Intuitively, it seems like the likely behavior will be that the computed GP "posterior" is flawed in a few ways: (1) it does not take into account the uncertainty over which actions are optimal; (2) it doesn't account for any optimism bias due to action selection; (3) the error bars are wrong because we're fitting non-Gaussian data using Gaussian tails; (4) maybe other problems I don't see yet. But my confidence in this description is low without further analysis and/or experiments. (And it's taken me, an expert reader, a fair bit of effort to come to this point -- so it's not great to ask every reader to spend this much effort or to have the necessary expertise to come to the same conclusion.)

---

### Author Response · Authors · 2021-11-23
**Common Comment For All Reviewers**

Dear Reviewers,

Thank you for your engagement with our work! Your reviews and comments have lead to major revisions in our work which we believe have significantly improved the quality and clarity of our work.

**Key Updates to the Manuscript**

* The section "Independence in Ensembles Matters" has undergone significant revisions:
** Section 4.1: We have replaced the old theorems (and accompanying text) with a new one directly comparing "Independent" and "Shared-LCB" ensembles.
** Section 4.2: We have updated the text, and added additional related experiments in this setting in Appendix G.2.
* We have added appendix H discussing the statistical model underlying our approach and the type of uncertainty we are aiming at.
* Experiments:
** We have included maze2d results alongside antmaze results in Table 1. maze2d is a simplified variant of antmaze where the agent is a pointmass instead of the ant agent. As can be seen in appendix Table 2 of the D4RL paper (https://arxiv.org/abs/2004.07219), many prior works in offline RL fail to even solve this variant of the maze tasks. Given that mazes represent the ability to stitch together different segments of different trajectories in an offline dataset, we believe our empirical results demonstrate that in terms of performance, MSG is a clear step up from prior work.
** We have added new experiments with the RL Unplugged benchmark [1] in Section 5.2.
** We have added Figure 2 as a means of better presenting the results of the very large Table 4 in the appendix.
* We have added Appendix I discussing practical hyperparameter tuning advice as well as presenting plots demonstrating the robustness of MSG with respect to hyperparameters.
* Overall we have fixed some readability issues and missing citations.

**Contributions and Novelties of Our Work**

* Estimating uncertainties using deep neural networks is a challenging and still open problem. It may be for this reason that not many works exist that combine offline RL + deep networks + uncertainty estimation (as we discussed in this response https://openreview.net/forum?id=wQ7RCayXUSl&noteId=T1pzSjUgBmj). A key contribution of our work is that we aimed to transfer the best methods for uncertainty estimation from supervised deep learning literature to offline RL with neural networks.
* "Deep Ensembles" are currently the state-of-the-art in supervised learning. To transfer their success to offline RL we showed 1) theoretically, 2) with toy experiments, and 3) with solid benchmark empirical results, that the common form of ensembling used in the literature, i.e. Shared-LCB or Shared-Min, are not nearly as effective as Independent ensembles, despite their implementation differing in only 2 lines of code.
* In addition to deep ensembles which are considered to be the state-of-the-art (SoTA) uncertainty estimation technique in the deep learning literature, we explored modern close to SoTA “efficient ensembles” and demonstrated their clear shortcoming when transferred to an RL setting, thus opening important avenues for continued research. Furthermore, we hope our significant empirical gains increases engagement from the community of deep network uncertainty estimation researchers who use supervised learning as the testbed for their approaches.
* The empirical results of MSG with deep ensembles is undeniably superior to prior work, as far as they can be captured by current benchmarks. No prior work has been able to make nearly as much progress on the maze2d and antmaze domains as MSG. Additionally, in our updated manuscript we have included results with the RL Unplugged benchmark [5]. Using network architectures that have $\frac{1}{60}$ of the parameters as typically used in RL Unplugged, MSG rivals the best reported results on these domains (with the exception of humanoid run task, which through behavioral cloning experiments we observed really does require the exceedingly large network sizes).

We hope that the contributions of our work and the updates we have made are appreciated by the reviewers. Thank you very much for an engaging Author Discussion Period!

Paper4539 Authors

[1] Gulcehre, Caglar, et al. "RL Unplugged: A Suite of Benchmarks for Offline Reinforcement Learning." Advances in Neural Information Processing Systems 33 (2020): 7248-7259.

---

### Decision · Program_Chairs · 2022-01-20

**Decision:**

Reject

**Comment:**

The paper attacks an interesting problem: accurately estimating uncertainties in action-value estimates in offline RL. It proposes a method based on ensembles of Q functions, where we alternately train an ensemble to estimate Q(s,a) for the current policy, and then adjust our policy based on the mean and uncertainty in this ensemble. By choosing mean + \beta * [standard deviation] as the basis for our policy updates, we can be either conservative (\beta < 0) or optimistic (beta > 0). The paper analyzes the ensemble training using the Gaussian process (NTK) view of deep nets.

The largest weakness of the paper is a lack of rigor in its analysis. While its main topic is uncertainty in Q estimates, the paper does not specify a valid probabilistic model on which such uncertainty estimates could be based. The theorems analyze only a part of the algorithm (policy evaluation), and don't take into account the interplay between this evaluation and any policy updates. The theorems also do not show that the computed output distribution is relevant to the actual uncertainty of the algorithm; e.g., they do not describe a prior for which the ensemble approximates the correct posterior (nor any other similar notion). Despite these omissions, the theorems are nonetheless presented as providing a reason to trust the output of the algorithm.

On the other hand, there definitely is valuable material in the paper; the experiments are interesting (and would be even more interesting if we could compare to some notion of a correct answer for at least the small ones), and the intuition and analysis could be enlightening if presented more clearly and formally, with a better description of the connection between theory and practice. Unfortunately, the paper as written doesn't enable the reader to accurately understand and assess the contributions.